# Contrasting with Symile: Simple Model-Agnostic Representation Learning for Unlimited Modalities

**Adriel Saporta**[*]    **Aahlad Puli**    **Mark Goldstein**    **Rajesh Ranganath**

New York University

## Abstract

Contrastive learning methods, such as CLIP, leverage naturally paired data—for example, images and their corresponding text captions—to learn general representations that transfer efficiently to downstream tasks. While such approaches are generally applied to two modalities, domains such as robotics, healthcare, and video need to support many types of data at once. We show that the pairwise application of CLIP fails to capture joint information between modalities, thereby limiting the quality of the learned representations. To address this issue, we present Symile, a simple contrastive learning approach that captures higher-order information between any number of modalities. Symile provides a flexible, architecture-agnostic objective for learning modality-specific representations. To develop Symile's objective, we derive a lower bound on total correlation, and show that Symile representations for any set of modalities form a sufficient statistic for predicting the remaining modalities. Symile outperforms pairwise CLIP, even with modalities missing in the data, on cross-modal classification and retrieval across several experiments including on an original multilingual dataset of 33M image, text and audio samples and a clinical dataset of chest X-rays, electrocardiograms, and laboratory measurements. All datasets and code used in this work are publicly available at https://github.com/rajesh-lab/symile.

## 1 Introduction

Contrastive learning leverages naturally paired data to learn general representations that transfer efficiently to downstream tasks [3, 35, 53]. A common contrastive approach is to maximize the mutual information between the paired modalities, ensuring that the learned representations retain sensitivity to all correlations between them. While SimCLR [12] popularized the use of the mutual information estimator InfoNCE [38] for data augmentations, CLIP [40] applied the approach to distinct modalities—for example, images and their corresponding text captions—where representations are learned using any encoder for each modality.

While contrastive approaches are generally applied to two modalities, there is a rapidly expanding range of domains that require the integration of many types of data at once. For example, in robotics, agents combine information from visual, proprioceptive, and tactile sensors [18, 28]; healthcare providers analyze various types of patient data including imaging, biosignals, and genomics [10, 29]; and video encompasses RGB frames, audio waveforms, and text transcripts [55]. One strategy for handling multimodal data has been to design specialized architectures capable of processing all data types at once, which limits their general applicability and increases operational complexity [2, 47]. Another common approach is to apply two-modality contrastive objectives, such as CLIP, to pairs of available modalities [15, 44].

In this paper, we show that, despite its popularity, the pairwise application of CLIP fails to capture higher-order conditional information between modalities, thereby limiting the quality of the

---

[*]Correspondence to: Adriel Saporta <adriel@nyu.edu>.

38th Conference on Neural Information Processing Systems (NeurIPS 2024).

representations it learns. For instance, given three modalities **a**, **b**, and **c**, pairwise CLIP captures dependencies between **a** and **b**, **b** and **c**, and **a** and **c**, yet cannot capture any conditional dependencies, such as between **a** and **b** *given* **c**. We show in Section 2.2 that even in a simple one-dimensional controlled setting where the target **b** is perfectly predictable from **a** and **c**, CLIP performs no better than random chance. Effective contrastive learning for more than two modalities requires a model-agnostic approach capable of learning modality-specific representations—like CLIP—yet also captures higher-order information between *any* number of modalities—unlike CLIP.

**Methodological contributions.** This paper presents *Symile*, a simple contrastive learning approach that captures higher-order information between any number of modalities. Symile provides a flexible, architecture-agnostic objective for learning modality-specific representations. To develop Symile's objective, we derive a total correlation estimator, employing a generalization of inner products to more than two vectors that allows for the simultaneous contrasting of all modalities and enables zero-shot applications such as classification and retrieval. We then show that the representations produced by Symile for any set of modalities form a sufficient statistic for predicting the remaining modalities not considered in the set. Because it targets total correlation, Symile captures *strictly more* information than CLIP, guaranteeing performance that matches or surpasses CLIP, except in cases where it known that *only* pairwise statistics are relevant. Given that such prior knowledge is rarely available, Symile should be favored over CLIP.

**Empirical contributions.** We demonstrate that Symile outperforms pairwise CLIP on cross-modal classification and retrieval across several experiments including on a multilingual dataset of images, text and audio of over 33M examples and a clinical dataset of chest X-rays, electrocardiograms, and laboratory measurements. We show that Symile retains its advantage over pairwise CLIP even with modalities missing in the data. We publicly release both the multilingual and the clinical datasets, which are specifically designed to test a model's ability to capture higher-order information between three distinct high-dimensional data types.

## 2 Background and motivation

In this section, we first provide background on the original CLIP objective for two modalities, and describe how it has been extended to additional modalities. We then present a simple problem set up for three modalities that illustrates where pairwise contrastive objectives fall short.

### 2.1 Pairwise contrastive learning

Given a batch of $(\mathbf{x}, \mathbf{y})$ pairs, separately encoded by $f_{\mathbf{x}}^{\boldsymbol{\theta}}$ and $f_{\mathbf{y}}^{\boldsymbol{\theta}}$, respectively, contrastive objectives such as CLIP maximize the similarity between representations of correctly paired (*positive*) samples and minimize the similarity between representations of incorrectly paired (*negative*) samples.

As is now standard in contrastive learning, in order to construct a batch of data, each modality is treated as the anchor in turn and used to construct a set of positive and negative samples. Letting $\tau \in \mathbb{R}^+$ be a temperature parameter, the CLIP objective when $\mathbf{x}$ is the anchor modality is the categorical cross-entropy of correctly classifying the positive pair out of $N$ possible pairs:

$$\ell^{(\mathbf{x} \to \mathbf{y})}(\boldsymbol{\theta}, \tau) = -\frac{1}{N} \sum_{i=1}^{N} \log \frac{\exp\left[\left(f_{\mathbf{x}}^{\boldsymbol{\theta}}(\mathbf{x}_i)^\top f_{\mathbf{y}}^{\boldsymbol{\theta}}(\mathbf{y}_i)\right)/\tau\right]}{\sum_{j=1}^{N} \exp\left[\left(f_{\mathbf{x}}^{\boldsymbol{\theta}}(\mathbf{x}_i)^\top f_{\mathbf{y}}^{\boldsymbol{\theta}}(\mathbf{y}_j)\right)/\tau\right]}. \tag{1}$$

The final CLIP objective is an average of the losses in each direction: $\mathcal{L}_{\text{CLIP}}^{(\mathbf{x},\mathbf{y})}(\boldsymbol{\theta}, \tau) = \frac{1}{2}\left[\ell^{(\mathbf{x} \to \mathbf{y})}(\boldsymbol{\theta}, \tau) + \ell^{(\mathbf{y} \to \mathbf{x})}(\boldsymbol{\theta}, \tau)\right]$. The dot product in Equation (1) serves as a scoring function that is trained to assign high values to positive pairs, which are sampled from the joint distribution $p_{\mathbf{x},\mathbf{y}}$, and low values to negative pairs, which are sampled from the product of marginals $p_{\mathbf{x}} p_{\mathbf{y}}$.

Contrastive methods are typically designed to maximize the mutual information between $\mathbf{x}$ and $\mathbf{y}$, which is defined as the Kullback–Leibler divergence from the joint distribution to the product of the marginal distributions: $\mathbf{I}(\mathbf{x}; \mathbf{y}) = D_{\text{KL}}\big(p(\mathbf{x}, \mathbf{y}) \parallel p(\mathbf{x})p(\mathbf{y})\big)$. It has been shown that Equation (1) maximizes a lower bound on the mutual information between $\mathbf{x}$ and $\mathbf{y}$ [38, 39]. This information maximization ensures that the learned representations preserve all correlations between the modalities, which is essential for downstream tasks.

**Incorporating additional modalities.** In order to learn a joint embedding space for more than two modalities, existing work has applied the CLIP objective in a pairwise fashion [1, 2, 9, 11, 14, 21,

33, 34, 43, 44, 47, 52]. For example, Guzhov et al. [19] extend CLIP to incorporate audio alongside image and text, and ImageBind [15] uses CLIP to align image embeddings with embeddings from five other modalities. In the simplest case, for three modalities, the pairwise CLIP loss corresponds to

$$\mathcal{L}_{\text{CLIP}}^{(\mathbf{x},\mathbf{y},\mathbf{z})}(\boldsymbol{\theta}, \tau) = \mathcal{L}_{\text{CLIP}}^{(\mathbf{x},\mathbf{y})}(\boldsymbol{\theta}, \tau) + \mathcal{L}_{\text{CLIP}}^{(\mathbf{y},\mathbf{z})}(\boldsymbol{\theta}, \tau) + \mathcal{L}_{\text{CLIP}}^{(\mathbf{x},\mathbf{z})}(\boldsymbol{\theta}, \tau).$$

CLIP can either be fine-tuned for downstream tasks or operate as a zero-shot classifier by computing the similarities between the *query* embedding from one modality and each *candidate* embedding from the other modality. In the case of more than two modalities, this generalizes to a sum across the pairwise similarities. The resulting similarity scores are used to rank the candidates, and the candidate with the highest similarity to the query is chosen [40].

## 2.2 A simple one-dimensional problem for three binary modalities

While contrastive objectives were originally designed for two modalities, the naive pairwise extension of CLIP to additional modalities warrants a deeper analysis. To explore this further, we propose a simple problem setup for the following data generating process:

$$\mathbf{a}, \mathbf{b} \sim \text{Bernoulli}(0.5), \quad \mathbf{c} = \mathbf{a} \text{ XOR } \mathbf{b}.$$

Using the pairwise CLIP objective, we fit three affine linear models to perform the zero-shot classification task of predicting whether $\mathbf{b}$ is 0 or 1 given $\mathbf{a}, \mathbf{c}$. See Appendix I for additional details.

Even in this simple one-dimensional controlled setting where the target $\mathbf{b}$ is perfectly predictable from $\mathbf{a}$ and $\mathbf{c}$, CLIP performs no better than random chance, with an accuracy of 0.5.

**CLIP failure analysis.** It can be shown that even though the variables $\mathbf{a}, \mathbf{b}, \mathbf{c}$ are jointly *dependent*—since $\mathbf{c}$ is a deterministic function of $\mathbf{a}$ and $\mathbf{b}$—they are pairwise independent (Appendix A):

$$\mathbf{I}(\mathbf{a}; \mathbf{b}) = \mathbf{I}(\mathbf{b}; \mathbf{c}) = \mathbf{I}(\mathbf{a}; \mathbf{c}) = 0, \quad \mathbf{I}(\mathbf{a}; \mathbf{b} \,|\, \mathbf{c}) > 0.$$

This explains CLIP's poor performance for the above XOR experiment: the objective maximizes a lower bound on the mutual information between pairwise terms, and therefore was not designed to capture higher-order dependencies such as the dependence between $\mathbf{a}$ and $\mathbf{b}$ given $\mathbf{c}$.[2] Capturing conditional dependencies like this will require the formulation of a new contrastive learning objective.

## 3 Learning Symile representations

Instead of targeting the mutual information between pairs of modalities, we target the *total correlation* between any number of modalities, learning what we call Symile[3] representations.

Total correlation [50]—the higher-order generalization of mutual information—is defined as the Kullback–Leibler divergence from the joint distribution to the product of the marginal distributions:

$$\mathbf{TC}(\mathbf{x}_1, \ldots, \mathbf{x}_M) = D_{\text{KL}}\big(p(\mathbf{x}_1, \ldots, \mathbf{x}_M) \,\|\, p(\mathbf{x}_1) \cdots p(\mathbf{x}_M)\big).$$

In words, total correlation is a symmetric statistical measure that captures the amount of information shared in a set of random variables. A higher total correlation implies more dependency among the variables, and a total correlation of zero indicates that the variables are independent.

Total correlation can be decomposed into a summation of mutual information terms. For example, in the case of three random variables,

$$3 \cdot \underbrace{\mathbf{TC}(\mathbf{x}, \mathbf{y}, \mathbf{z})}_{\text{Symile target}} = \big[\mathbf{I}(\mathbf{x}; \mathbf{y}) + \mathbf{I}(\mathbf{z}; \mathbf{x}, \mathbf{y})\big] + \big[\mathbf{I}(\mathbf{y}; \mathbf{z}) + \mathbf{I}(\mathbf{x}; \mathbf{y}, \mathbf{z})\big] + \big[\mathbf{I}(\mathbf{x}; \mathbf{z}) + \mathbf{I}(\mathbf{y}; \mathbf{x}, \mathbf{z})\big]$$

$$= 2 \cdot \underbrace{\big[\mathbf{I}(\mathbf{x}; \mathbf{y}) + \mathbf{I}(\mathbf{y}; \mathbf{z}) + \mathbf{I}(\mathbf{x}; \mathbf{z})\big]}_{\substack{\text{pairwise information} \\ \text{(CLIP target)}}} + \underbrace{\mathbf{I}(\mathbf{x}; \mathbf{y} \,|\, \mathbf{z}) + \mathbf{I}(\mathbf{y}; \mathbf{z} \,|\, \mathbf{x}) + \mathbf{I}(\mathbf{x}; \mathbf{z} \,|\, \mathbf{y})}_{\text{higher-order information}}. \quad (2)$$

While, as discussed, contrastive learning was designed to capture the shared information between modalities, Equation (2) indicates that when there are more than two modalities, the scope of what to capture should extend beyond pairwise information to include conditional interactions (Figure 1).

---

[2]To be specific, we use "higher-order information" to mean information between two random variables given any number of additional random variables in the conditioning set.

[3]Symile stands for SYmmetric MultILinear Embeddings.

Because it targets total correlation, Symile captures *strictly more* information than CLIP, guaranteeing performance that matches or surpasses CLIP, except in cases where only pairwise statistics are relevant, with no higher-order interactions whatsoever. In such cases, Symile may be less sample efficient, as it tracks both pairwise and higher-order information. Unless there is prior knowledge that the downstream task relies *solely* on pairwise statistics, Symile should be chosen over CLIP.

To illustrate when such higher-order information might be relevant, consider again the XOR experiment outlined in Section 2.2. Because all the pairwise information terms between $\mathbf{a}$, $\mathbf{b}$, and $\mathbf{c}$ are zero, the conditional mutual information terms constitute the only dependence between the variables to track.

The XOR experiment represents an extreme case where the CLIP target is zero, but most real-world applications will exhibit a combination of both pairwise and higher-order information. For example, in order to diagnose acute pancreatitis, one might consider a patient's clinical history of abdominal pain, elevated levels of digestive enzymes, and imaging results consistent with

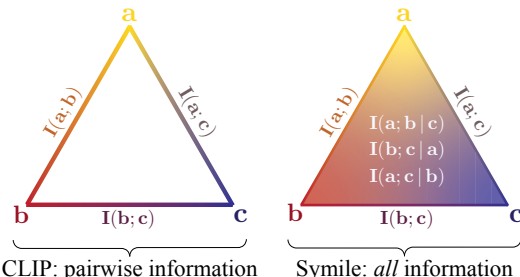

Figure 1: An illustrative comparison of the information captured by CLIP (only pairwise) and Symile (both pairwise and higher-order).

inflammation. While each of these modalities would provide useful information about the likelihood of pancreatitis (i.e., pairwise information between the modality and the diagnosis is non-zero), none of them alone would be diagnostic of the condition. Similarly, in the case of Parkinson's disease, clinical evaluation provides valuable information, along with imaging and blood tests to rule out other conditions, but clinicians rely on the integration of all modalities.

### 3.1 Deriving a multi-sample lower bound on total correlation

In order to eventually derive a contrastive objective by maximizing total correlation, we first establish a multi-sample lower bound on total correlation. This lower bound and, in the next section, the Symile objective are illustrated using three modalities for simplicity, but both can be extended to an arbitrary number of modalities, as shown in Appendix B.

Given a batch of $N$ $(\mathbf{x}, \mathbf{y}, \mathbf{z})$ triples, let

$$\mathbf{i} \sim \text{Uniform}(\{1, \ldots, N\}) \tag{3}$$

denote the index of the positive triple in the batch. Our goal is to estimate $\mathbf{TC}(\mathbf{x}, \mathbf{y}, \mathbf{z})$ given one positive triple sampled from the joint distribution, and $N - 1$ negative triples sampled from the product of marginals:

$$\mathbf{x}, \mathbf{y}_i, \mathbf{z}_i \sim p_{\mathbf{x}, \mathbf{y}, \mathbf{z}}(\mathbf{x}, \mathbf{y}_i, \mathbf{z}_i), \quad \mathbf{x}, \mathbf{y}_{j \neq i}, \mathbf{z}_{j \neq i} \sim p(\mathbf{x}) p_{\mathbf{y}}(\mathbf{y}_{j \neq i}) p_{\mathbf{z}}(\mathbf{z}_{j \neq i}).$$

Letting $\mathbf{Y}_N = \{\mathbf{y}_n\}_{n=1}^N$ and $\mathbf{Z}_N = \{\mathbf{z}_n\}_{n=1}^N$ be the sets of all samples of $\mathbf{y}$ and $\mathbf{z}$, respectively, this sampling procedure describes the following distribution:

$$p(\mathbf{x}, \mathbf{Y}_N, \mathbf{Z}_N \mid \mathbf{i} = i) = p(\mathbf{x}) \overbrace{p_{\mathbf{y}, \mathbf{z} \mid \mathbf{x}}(\mathbf{y}_i, \mathbf{z}_i \mid \mathbf{x})}^{\substack{\mathbf{y}, \mathbf{z} \text{ from} \\ \text{positive sample}}} \overbrace{\left[ \prod_{j \neq i} p_{\mathbf{y}}(\mathbf{y}_j) \right] \left[ \prod_{j \neq i} p_{\mathbf{z}}(\mathbf{z}_j) \right]}^{\substack{\mathbf{y}, \mathbf{z} \text{ from} \\ \text{negative samples}}}. \tag{4}$$

We derive the following lower bound in Appendix B:

**Theorem 3.1** (Total Correlation Lower Bound). *Given the distributions in Equations* (3) *and* (4), *for any value $i$ of $\mathbf{i}$ and any scoring function $g$, a multi-sample contrastive lower bound on total correlation is*

$$\mathbf{TC}(\mathbf{x}, \mathbf{y}, \mathbf{z}) \geq \log N + \underset{p(\mathbf{x}, \mathbf{Y}_N, \mathbf{Z}_N \mid \mathbf{i} = i)}{\mathbb{E}} \log \frac{\exp g(\mathbf{x}, \mathbf{y}_i, \mathbf{z}_i)}{\sum_{j=1}^N \exp g(\mathbf{x}, \mathbf{y}_j, \mathbf{z}_j)}. \tag{5}$$

As described in Section 2.1, in contrastive learning each modality is sequentially treated as the anchor, with a batch of corresponding positive and negative samples generated for each. Theorem 3.1 treats $\mathbf{x}$ as the anchor modality, but by symmetry holds when $\mathbf{y}$ or $\mathbf{z}$ acts as the anchor modality.

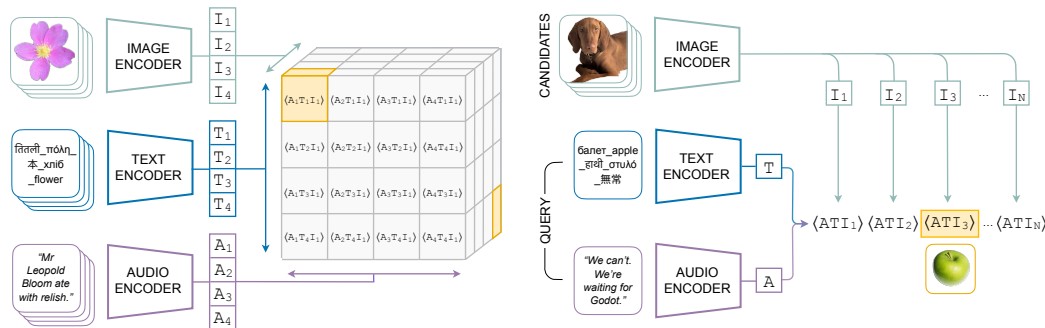

(a) Symile pre-training  (b) Zero-shot prediction

Figure 2: Symile pre-training and zero-shot prediction on the Symile-M3 multilingual dataset. (a) Given a batch of triples, Symile maximizes the multilinear inner product (MIP) of positive triples (in yellow along the diagonal of the cube) and minimizes the MIP of negative triples. (b) The model selects the candidate image with the highest similarity to the query audio and text.

Notice that the term inside the expectation in Equation (5) is the categorical log likelihood of correctly identifying the index of the positive triple in the batch, where the *scoring function* (or *critic*) $g$ is trained to assign a high value to positive samples and a low value to negative samples. In Appendix E, we show that the optimal scoring function $g^*$ is equal to the instantaneous total correlation up to additive constants:

**Lemma 3.2.** *For some $\kappa > 0$, the g that maximizes the lower bound*

$$\mathbf{TC}(\mathbf{x}, \mathbf{y}, \mathbf{z}) \geq \log N + \underset{p(\mathbf{x}, \mathbf{Y}_N, \mathbf{Z}_N \mid \mathbf{i}=i)}{\mathbb{E}} \log \frac{\exp g(\mathbf{x}, \mathbf{y}_i, \mathbf{z}_i)}{\sum_{j=1}^{N} \exp g(\mathbf{x}, \mathbf{y}_j, \mathbf{z}_j)}$$

*is*

$$g^*(\mathbf{x}, \mathbf{y}, \mathbf{z}) = \kappa + \log \left[ \frac{p_{\mathbf{x}, \mathbf{y}, \mathbf{z}}(\mathbf{x}, \mathbf{y}, \mathbf{z})}{p(\mathbf{x}) p_{\mathbf{y}}(\mathbf{y}) p_{\mathbf{z}}(\mathbf{z})} \right].$$

We show in Appendix B.3 that, as $N$ gets larger, the total correlation lower bound closes for the optimal scoring function $g^*$. This implies a computational-statistical trade-off: a larger batch size demands more computation but results in a tighter bound.

### 3.2 The Symile objective

We now derive the Symile loss by maximizing the total correlation lower bound in Theorem 3.1.

Instead of using the dot product as a scoring function, as CLIP does, Symile uses its generalized form: the coordinate-wise sum of the element-wise product of a set of vectors. We call this the multilinear inner product (MIP): $\langle \{\mathbf{x}_i\}_{i=1}^{M} \rangle = \sum_{d=1}^{D} \prod_{i=1}^{M} x_{i,d}$. As a scoring function, the MIP strikes a balance between computational simplicity and expressive power: it represents one of the simplest possible generalizations of the dot product to more than two modalities, and the vector multiplication ensures it is expressive enough to model any joint statistic.[4]

Given a batch of $N'$ positive triples $(\mathbf{x}_i, \mathbf{y}_i, \mathbf{z}_i)$, each with $N - 1$ corresponding negative triples $(\mathbf{x}_i, \mathbf{y}'_j, \mathbf{z}'_j)$, and letting $\tau \in \mathbb{R}^+$ be a temperature parameter, the Symile loss is the negative of an empirical estimate of the expected log likelihood in Equation (5):

$$\ell^{(\mathbf{x} \rightarrow \mathbf{y}, \mathbf{z})}(\boldsymbol{\theta}, \tau) =$$

$$- \frac{1}{N'} \sum_{i=1}^{N'} \log \frac{\exp\left(\langle f_{\mathbf{x}}^{\boldsymbol{\theta}}(\mathbf{x}_i), f_{\mathbf{y}}^{\boldsymbol{\theta}}(\mathbf{y}_i), f_{\mathbf{z}}^{\boldsymbol{\theta}}(\mathbf{z}_i)\rangle / \tau\right)}{\exp\left(\langle f_{\mathbf{x}}^{\boldsymbol{\theta}}(\mathbf{x}_i), f_{\mathbf{y}}^{\boldsymbol{\theta}}(\mathbf{y}_i), f_{\mathbf{z}}^{\boldsymbol{\theta}}(\mathbf{z}_i)\rangle / \tau\right) + \sum_{j=1}^{N-1} \exp\left(\langle f_{\mathbf{x}}^{\boldsymbol{\theta}}(\mathbf{x}_i), f_{\mathbf{y}}^{\boldsymbol{\theta}}(\mathbf{y}'_j), f_{\mathbf{z}}^{\boldsymbol{\theta}}(\mathbf{z}'_j)\rangle / \tau\right)}. \quad (6)$$

---

[4]Note that the MIP is a measure of similarity defined by the joint distribution of the modalities, rather than a measure of the *geometric* similarity of the modalities' representations. For example, a large MIP for Symile representations $\mathbf{r}_{\mathbf{x}}, \mathbf{r}_{\mathbf{y}}, \mathbf{r}_{\mathbf{z}}$ indicates that the sample $(\mathbf{x}, \mathbf{y}, \mathbf{z})$ has high probability under the joint likelihood; it provides no information about whether $\mathbf{r}_{\mathbf{x}}, \mathbf{r}_{\mathbf{y}}, \mathbf{r}_{\mathbf{z}}$ are equal to one another.

Minimizing Equation (6) optimizes the lower bound on total correlation by maximizing the MIP of positive tuples and minimizing the MIP of negative tuples (Figure 2a). See Appendix B.4 for the Symile objective generalized to any number of modalities.

As is done with CLIP, the final Symile loss is an average of the loss terms where each modality is treated as the anchor in turn:

$$\mathcal{L}_{\text{Symile}}^{(\mathbf{x},\mathbf{y},\mathbf{z})}(\boldsymbol{\theta}, \tau) = \frac{1}{3}\big[\ell^{(\mathbf{x}\to\mathbf{y},\mathbf{z})}(\boldsymbol{\theta}, \tau) + \ell^{(\mathbf{y}\to\mathbf{x},\mathbf{z})}(\boldsymbol{\theta}, \tau) + \ell^{(\mathbf{z}\to\mathbf{x},\mathbf{y})}(\boldsymbol{\theta}, \tau)\big].$$

**Efficient negative sampling.** In the sampling procedure described in Section 3.1, negatives samples for the non-anchor modalities are drawn independently for each positive triple, which can be intensive in terms of both computation and memory. Instead, for efficiency, negative sampling can be approximated within a batch by forming negative tuples from non-matching combinations of the non-anchor modalities.

Approximating negatives within a batch is straightforward with two modalities, but in the case of more than two modalities, both how negatives are formed and how many are used become design choices. At one extreme, one could generate $N^2 - 1$ negative triples for each positive by considering all possible combinations of the two remaining non-anchor modalities. This approach, which we call $O(N^2)$, can be compu-

**Algorithm 1** Pseudocode for implementation of Symile with $O(N)$ negative sampling

```
# compute [n, n] logits from x → (y, z)
def get_logits(x, y, z):
  MIP_pos = (x * y * z).sum(axis=1)  #[n]
  y_shuffled = y[np.random.permutation(n)]
  z_shuffled = z[np.random.permutation(n)]
  MIP_neg = x @ (y_shuffled * z_shuffled).T  #[n, n]
  return np.where(np.eye(n), MIP_pos, MIP_neg)

# v, u, w:  L2-normalized embeddings, each [n, dim]
def symile_loss(v, u, w):
  logits_v_uw = np.exp(t) * get_logits(v, u, w)
  logits_u_vw = np.exp(t) * get_logits(u, v, w)
  logits_w_vu = np.exp(t) * get_logits(w, v, u)
  labels = np.arange(n)
  loss_v_uw = ce_loss(logits_v_uw, labels)
  loss_u_vw = ce_loss(logits_u_vw, labels)
  loss_w_vu = ce_loss(logits_w_vu, labels)
  return (loss_v_uw + loss_u_vw + loss_w_vu)/3
```

tationally and memory intensive. Instead, any subset of these negatives can be used for sampling. For instance, a more efficient approach, which we refer to as $O(N)$, involves randomly permuting the non-anchor modalities within the batch, providing each data point with $N - 1$ negatives. The cube in Figure 2a illustrates the $O(N^2)$ approach and Algorithm 1 presents pseudocode for the $O(N)$ approach, both for three modalities.

**Missing data.** The Symile objective is defined for data in which all modalities are observed. However, in practice, datasets often include samples where not all modalities are available. This raises the question: during training how should one incorporate data points for which only a subset of modalities is observed? Symile can be easily adapted to such missingness by adding extra dimensions to the encoder inputs that indicate whether or not a modality is missing, ensuring that missing data points are out-of-support. This approach allows Symile to model dependencies between whichever modalities are observed within a sample. We show in Section 5.2 that Symile retains its advantage over pairwise CLIP even with modalities missing in the data.

### 3.3 Learning sufficient statistics with Symile

An important property of Symile is that it learns sufficient statistics, which is central to the representations' effectiveness for downstream tasks.

**Theorem 3.3** (Symile Sufficient Statistics). *Let* $\mathbf{x}, \mathbf{y}, \mathbf{z}$ *be three random variables whose optimal representations when trained using Symile are* $f_{\mathbf{x}}^*(\mathbf{x}), f_{\mathbf{y}}^*(\mathbf{y}), f_{\mathbf{z}}^*(\mathbf{z})$, *respectively. The element-wise product of any subset of the representations is a sufficient statistic for predicting the remaining random variables.*

*For example,* $f_{\mathbf{x}}^*(\mathbf{x}) \odot f_{\mathbf{z}}^*(\mathbf{z})$ *is a sufficient statistic for predicting* $\mathbf{y}$, *which can be expressed using the following conditional independence statement:*

$$\mathbf{y} \perp\!\!\!\perp \mathbf{x}, \mathbf{z} \mid f_{\mathbf{x}}^*(\mathbf{x}) \odot f_{\mathbf{z}}^*(\mathbf{z}).$$

The proof can be found in Appendix G. The independence statement in Theorem 3.3 tells us that the element-wise product of the Symile representations of any subset of modalities contains all the information required to predict the remaining modalities. In other words, once Symile representations have been computed, access to the full data is no longer needed. Theorem 3.3 confirms Symile's ability to learn efficient modality-specific representations for downstream tasks.

### 3.4 Zero-shot prediction using the scoring function

Just as with CLIP, the optimal scoring function $g^*$ (Lemma 3.2) can be used to predict one of the modalities $y \in \mathcal{Y}$ using instances of the other modalities $x, z$. If $p(\mathbf{y})$ is uniformly distributed, then the scoring function can be used to rank the candidates for $\mathbf{y}$: $\arg\max_{y \in \mathcal{Y}} p(\mathbf{y} = y \mid x, z) = \arg\max_{y \in \mathcal{Y}} g^*(x, y, z)$.

However, this zero-shot approach, whether applied to Symile or to CLIP, does not lead to the Bayes optimal prediction and, consequently, does not always yield reliable results when $p(\mathbf{y})$ is *not* uniformly distributed (see Appendix H for a detailed discussion). To address this issue, we can instead compute the desired conditional probability directly using the scoring function:

**Theorem 3.4** (Conditional Distribution using the Scoring Function). *Let* $\mathbf{x}, \mathbf{y}, \mathbf{z}$ *be three random variables whose optimal representations when trained using Symile are* $f_{\mathbf{x}}^*(\mathbf{x}), f_{\mathbf{y}}^*(\mathbf{y}), f_{\mathbf{z}}^*(\mathbf{z})$, *respectively. Let the* MIP $\langle f_{\mathbf{x}}^*(\mathbf{x}), f_{\mathbf{y}}^*(\mathbf{y}), f_{\mathbf{z}}^*(\mathbf{z}) \rangle$ *be the scoring function. Then,*

$$p(\mathbf{y} \mid \mathbf{x}, \mathbf{z}) = \frac{\exp\left[\langle f_{\mathbf{x}}^*(\mathbf{x}), f_{\mathbf{y}}^*(\mathbf{y}), f_{\mathbf{z}}^*(\mathbf{z}) \rangle\right] p(\mathbf{y})}{\int_{\mathbf{y}} \exp\left[\langle f_{\mathbf{x}}^*(\mathbf{x}), f_{\mathbf{y}}^*(\mathbf{y}), f_{\mathbf{z}}^*(\mathbf{z}) \rangle\right] p(\mathbf{y}) d\mathbf{y}}. \tag{7}$$

The proof is provided in Appendix H.

If the marginal distribution of $\mathbf{y}$ is known, we could then perform zero-shot classification in one of two ways. When the distribution $p(\mathbf{y} \mid \mathbf{x}, \mathbf{z})$ itself is of interest, as is often the case in healthcare [10], we could compute $p(\mathbf{y} \mid \mathbf{x}, \mathbf{z})$ directly, following Equation (7). Alternatively, if only predictions are needed, we could use

$$\langle f_{\mathbf{x}}^*(\mathbf{x}), f_{\mathbf{y}}^*(\mathbf{y}), f_{\mathbf{z}}^*(\mathbf{z}) \rangle + \log p(\mathbf{y})$$

to rank the possible values for $\mathbf{y}$, as discussed further in Appendix H. If the marginal distribution of $\mathbf{y}$ is *not* known, then because $f_{\mathbf{x}}^*(\mathbf{x}) \odot f_{\mathbf{z}}^*(\mathbf{z})$ is a sufficient statistic for predicting $\mathbf{y}$ (Theorem 3.3), we could instead use $f_{\mathbf{x}}^*(\mathbf{x}) \odot f_{\mathbf{z}}^*(\mathbf{z})$ to train a simple model to predict any property of $\mathbf{y}$, $s(\mathbf{y})$: $p(s(\mathbf{y}) \mid f_{\mathbf{x}}^*(\mathbf{x}) \odot f_{\mathbf{z}}^*(\mathbf{z}))$.

Note that although the above discussion centers on Symile, it applies equally to CLIP and its own scoring function, the dot product.

## 4 Related work

**Contrastive learning beyond two modalities.** As discussed, previous work has extended contrastive learning to multiple modalities by applying CLIP to pairs of available modalities. Tian et al. [49] distinguish between two such pairwise approaches: core view and full graph. The core view strategy fixes one modality and then averages the loss terms between that primary modality and each of the other modalities [1, 11, 44]. ImageBind [15] exemplifies this approach, using CLIP to align image embeddings with embeddings from five other modalities: text, audio, depth, thermal, and motion sensor data. One advantage of this strategy is that it avoids the need for datasets with all modalities (though each dataset must still align with a primary modality). As discussed in Sections 3.2 and 5.2, Symile representations can be learned even with modalities missing in the data.

The full graph strategy—which we have referred to as pairwise CLIP in this paper—is to consider all $\binom{M}{2}$ contrastive losses [9, 14, 33, 34, 43]. For example, Guzhov et al. [19] extend CLIP to include audio with text-to-image, text-to-audio, and image-to-audio losses. While this pairwise strategy captures strictly more information than the one used by ImageBind, neither pairwise approach is able to capture the higher-order information that Symile does.

Pairwise CLIP has also been applied to architecture-specific fusion models that simultaneously process modalities to capture cross-modal interactions [2, 21, 52]. For example, Shvetsova et al. [47] train a Transformer to accept any number of modalities, using a weighted sum of contrastive losses across all input combinations. Such fusion approaches face a combinatorial explosion not only in the number of weighting coefficients to tune, but also in the number of forward passes required per batch. In contrast, Symile is architecture-agnostic and can learn modality-specific representations.

**Targeting higher-order information with contrastive learning.** The use of contrastive methods to target higher-order information has been explored primarily within the context of multiple augmentations of the same data. For instance, Bai et al. [5] derive a total correlation estimator by

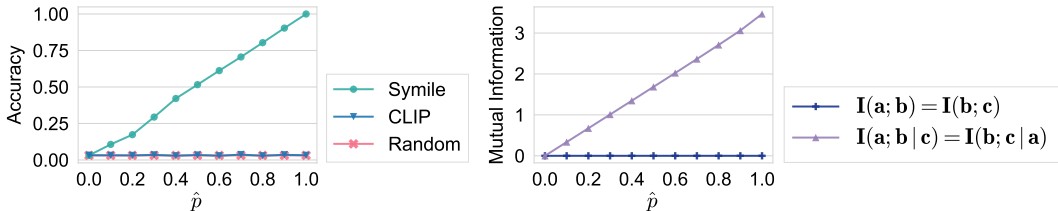

Figure 3: The performance gap between Symile and CLIP on binary synthetic data (left) is a consequence of the changing information dynamics between the variables as $\hat{p}$ moves from 0 to 1 (right). Mean accuracy is reported across 10 bootstrap samples of the test set.

recursively decomposing total correlation into a summation of mutual information terms, to which variational estimators are applied (in contrast, Symile optimizes only a single term when targeting total correlation). They then use their estimator to maximize the total correlation between four text augmentations. Shidani et al. [46] develop a pairwise contrastive approach for image representation learning by generalizing a lower bound on mutual information to one-vs-rest mutual information across multiple augmentations. Liang et al. [31] maximize the information in two modalities for a specific downstream task by targeting higher-order information.

The relationship between these studies and our work is analogous to that between SimCLR [12] and CLIP. SimCLR popularized the use of the InfoNCE mutual information estimator for contrastive learning on two data augmentations. Building on this framework, CLIP applied the approach to distinct modalities, where representations are learned separately for each modality using any encoder. Similarly, while existing work leverages total correlation or mutual information estimators for multi-augmentation contrastive learning, to our knowledge only pairwise applications of CLIP have applied such estimators to more than two distinct modalities. Our work parallels the contributions of InfoNCE and CLIP for cases involving more than two modalities: like InfoNCE, we develop a simple estimator that recovers all possible information between any number of modalities, and like CLIP, we show how this estimator can be used to learn modality-specific representations using any encoder.

## 5 Experiments

In this section, we empirically evaluate Symile on cross-modal retrieval tasks in three settings: a synthetic dataset, a multilingual dataset encompassing text, images, and audio, and a clinical dataset with chest X-rays, electrocardiograms, and blood labs. Throughout our experiments, we use pairwise CLIP as a baseline comparison since, as outlined in Section 4, it represents the only architecture-agnostic approach that applies contrastive objectives to more than two modalities. We release all datasets and code used in these experiments at https://github.com/rajesh-lab/symile.

### 5.1 Synthetic data

Building on the illustrative XOR experiment from Section 2, we first test Symile on a synthetic dataset drawn according to the following sampling procedure:

$$a_j, b_j \sim \text{Bernoulli}(0.5), \quad i \sim \text{Bernoulli}(\hat{p}), \quad c_j = (a_j \text{ XOR } b_j)^i \cdot a_j^{(1-i)}.$$

We fit three affine linear functions that map $\mathbf{a}, \mathbf{b}, \mathbf{c} \in \mathbb{R}^5$ to representations $\mathbf{r_a}, \mathbf{r_b}, \mathbf{r_c} \in \mathbb{R}^{16}$, respectively, and evaluate the model's ability to correctly predict $\mathbf{r_b}$ given the pair $(\mathbf{r_a}, \mathbf{r_c})$.

**Results.** Figure 3 (left) compares Symile and CLIP across varying values of $\hat{p}$. Both models start with a mean accuracy of $0.032 \pm 0.001$ (SE) at $\hat{p} = 0$. As $\hat{p}$ increases, Symile's accuracy progressively climbs, reaching perfect accuracy at $\hat{p} = 1 \pm 0.0$ (SE). In contrast, CLIP's accuracy remains nearly constant, barely surpassing the baseline random guessing rate of $0.031$ ($1/32$).

This performance gap is a consequence of the changing information dynamics between the variables as $\hat{p}$ moves from 0 to 1, as shown in Figure 3 (right). When $\hat{p} = 0$, $\mathbf{b}$ shares no information with $\mathbf{a}$ and $\mathbf{c}$—either pairwise or conditionally—rendering both models incapable of predicting $\mathbf{r_b}$ from $(\mathbf{r_a}, \mathbf{r_c})$. As $\hat{p}$ increases, the higher-order $\mathbf{I}(\mathbf{a}; \mathbf{b} \mid \mathbf{c})$ and $\mathbf{I}(\mathbf{c}; \mathbf{b} \mid \mathbf{a})$ rise, driving a corresponding improvement in Symile's performance. However, because the pairwise $\mathbf{I}(\mathbf{a}; \mathbf{b})$ and $\mathbf{I}(\mathbf{b}; \mathbf{c})$ are always zero, there is no value of $\hat{p}$ at which CLIP is able to predict $\mathbf{r_b}$ from $(\mathbf{r_a}, \mathbf{r_c})$.

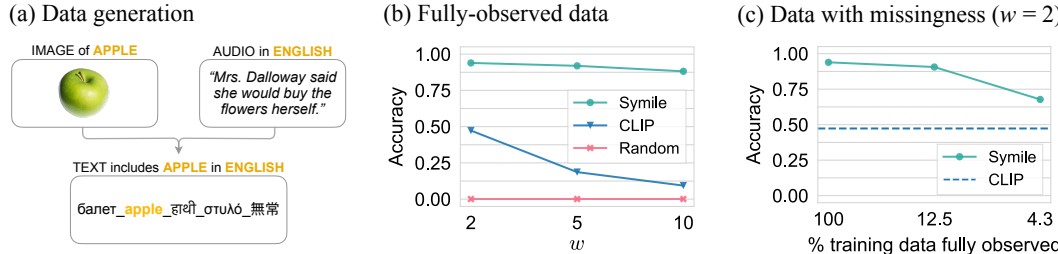

Figure 4: (a) Data-generating process for Symile-M3-5. (b) Comparison of Symile and CLIP on the three versions of Symile-M3 ($w \in \{2, 5, 10\}$). Random chance is $^1/_{1000}$. Symile successfully leverages joint information between the modalities, whereas CLIP is limited to pairwise information, resulting in accuracies bounded by $^1/_w$. (c) Symile outperforms the CLIP baseline on Symile-M3-2 across varying levels of completeness in the training data. Both plots report mean accuracy across 10 bootstrap samples of the test set.

### 5.2   Symile-M3: a multilingual dataset

We now evaluate Symile on a new multilingual dataset comprising 33 million (audio, image, text) samples. The dataset, Symile-M3, is specifically designed to test a model's ability to capture higher-order information between three distinct high-dimensional data types: by incorporating multiple languages, we construct a task where text and audio are both needed to predict the image, and where, importantly, neither text nor audio alone would suffice.

**Dataset design and model setup.**   Let $w$ represent the number of languages in the dataset. An (audio, image, text) sample is generated by first drawing a short one-sentence audio clip from Common Voice [4] spoken in one of $w$ languages with equal probability. An image is drawn from ImageNet [45] that corresponds to one of 1,000 classes with equal probability. Finally, text containing exactly $w$ words is generated based on the drawn audio and image: one of the $w$ words in the text is the drawn image class name in the drawn audio language. The remaining $w - 1$ words are randomly chosen from the ImageNet class names and written in one of the $w$ languages such that there is no overlap in language or class name across the $w$ words in the text. The words are separated by underscores, and their order is randomized. We release three versions of the dataset: Symile-M3-2, Symile-M3-5, and Symile-M3-10, corresponding to 2, 5, and 10 languages ($w$). Figure 4a shows an example of the data-generating process for Symile-M3-5. For each of the three datasets, 10M training, 500K validation, and 500K test samples were generated.

We use pre-trained encoders, freezing all parameters except for those in the text encoder's embedding layer and first encoder layer, which are fine-tuned. We train three linear projections to map each encoder's representation to the same 8192-dimensional space. The Symile loss is trained with $O(N)$ negative sampling. See Appendix I for details.

**Evaluation and results.**   We evaluate the learned representations on the zero-shot retrieval task of finding an image of the appropriate class given the audio and text. The most probable image for a given *query* audio and text pair, selected from all possible *candidate* images in the test set, is that with the highest similarity score (Figure 2b). Symile-M3 was designed to ensure that neither text nor audio alone would suffice to predict the image. Therefore, success on this zero-shot retrieval task hinges on a model's ability to capture joint information between the three modalities.

As shown in Figure 4b, Symile successfully leverages this joint information, with mean accuracies of 0.939, 0.919, and 0.882 on Symile-M3-2, Symile-M3-5, and Symile-M3-10, respectively, calculated across 10 bootstrap samples of the test set, all with standard error less than $4.0 \times 10^{-4}$. In contrast, CLIP, which captures pairwise information between image and text, can only predict an image randomly from among the $w$ class labels present in the text, resulting in mean accuracies of 0.473, 0.187, and 0.094 on Symile-M3-2, Symile-M3-5, and Symile-M3-10, respectively, all with standard error $\leq 3.01 \times 10^{-4}$. Because CLIP cannot distinguish between the class labels in the text using the audio language, it can only pick a class label at random, bounding its accuracy by $^1/_w$.

**Missing data.**   We also train Symile on a variant of Symile-M3-2 where each modality is independently missing with probability 0.5 or 0.65, corresponding, respectively, to probabilities 0.125 and 0.043 of a complete data sample in the training set (see Appendix I for details). As before, the test set consists of complete triples. As shown in Figure 4c, even when only 12.5% of the training data is

complete, Symile achieves a mean accuracy of $0.906 \pm 3.4 \times 10^{-4}$ (SE), far outperforming the CLIP baseline accuracy of $0.473$, despite the adverse effect of missing modalities. Notably, when less than $5\%$ of the training data is complete, Symile still exceeds the CLIP baseline.

### 5.3 Chest X-ray prediction using electrocardiograms and laboratory measurements

Zero-shot retrieval is widely used in the evaluation of representation learning for healthcare [6, 22, 29, 51, 56]. In this section, we evaluate the Symile objective on Symile-MIMIC, a clinical dataset comprised of chest X-rays, electrocardiograms, and blood labs from MIMIC-IV [17, 24, 27] and MIMIC-CXR [25, 26]. Since ECGs and labs are both safer than CXRs, this experiment explores whether an ECG and labs collected at admission are predictive of a CXR taken shortly thereafter.

**Dataset design and model setup.** Each data sample includes an ECG reading and blood labs taken within 24 hours of the patient's admission to the hospital, and a CXR taken in the 24- to 72-hour period post-admission (Figure 5a). Our analysis focuses on the 50 most common blood labs, with each sample containing at least one.

We split our dataset ($11,622$ admissions) into a train/validation development set ($95\%$ of patients) and a test set ($5\%$ of patients), ensuring there is no patient overlap across the splits. Following previous work, we use the ResNet-50 and ResNet-18 architectures [20] for the CXR and ECG encoders, respectively, and a three-layer neural network to encode the blood labs. All encoders are trained from scratch, and three linear projections map each encoder's representation to the same 8192-dimensional space. Given the limited size of the dataset, the Symile loss is trained with $O(N^2)$ negative sampling to mitigate overfitting. See Appendix I for details.

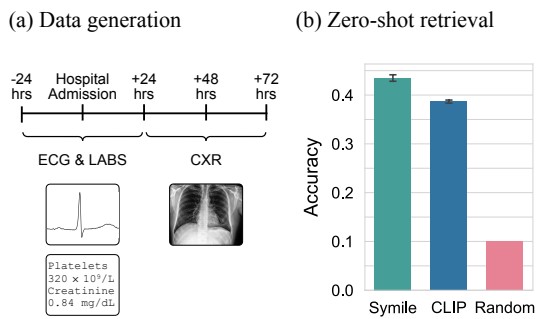

Figure 5: (a) Each sample of Symile-MIMIC includes an ECG and blood labs taken within 24 hours of the patient's admission to the hospital, and a CXR taken in the 24- to 72-hour period post-admission. (b) Retrieval accuracy for identifying the CXR corresponding to a given ECG and labs pair. Results are averaged over 10 bootstrap samples, with error bars indicating standard error.

**Evaluation and results.** We evaluate the learned representations on the zero-shot retrieval task of finding the most probable *candidate* CXR for a given *query* ECG and labs pair according to the similarity score. For each query ECG and labs pair in the test set, we sample nine negative CXR candidates from the remaining test samples, so that that each query has a total of 10 candidates: one positive (the true corresponding CXR) and nine negative.

In Figure 5b, we report mean accuracy for Symile and CLIP over 10 bootstrap samples of the test set. While both models surpass random chance (0.1), Symile achieves an average accuracy of $0.435 \pm 0.007$ (SE), outperforming CLIP's $0.387 \pm 0.003$ (SE). These results correspond to a $12.5\%$ increase in accuracy for Symile over CLIP.

## 6 Conclusion

This work presents Symile, a simple contrastive learning approach that captures higher-order information between any number of modalities. Symile provides a flexible, architecture-agnostic objective for learning modality-specific representations, maintaining the simplicity of CLIP while delivering superior performance, even in cases of missing modalities. Because it targets total correlation, Symile captures strictly more information than CLIP, guaranteeing performance that matches or surpasses CLIP, except in cases where it known that only pairwise statistics are relevant. Given that such prior knowledge is rarely available, Symile should be favored over CLIP.

**Future work.** (1) The sigmoid-based loss function SigLIP [54] was recently introduced as a memory-efficient alternative to traditional softmax-based contrastive objectives. A potential avenue for future work would be to adapt Symile, and its use of the multilinear inner product, to this sigmoid loss. (2) The proposed implementation of Symile relies on an approximation for negative sampling, and future work could examine how this approximation scales when applied to settings with more than three modalities. (3) Future work could integrate pre-trained Symile representations into multimodal large language models, enabling them to capture higher-order information between modalities.

# 7 Acknowledgements

We are especially grateful to Charley Crissman for his invaluable and meticulous feedback on every aspect of the paper, from the proofs to the code. We would like to thank Nick Murphy (Pantograph) and Madeleine Murphy for their thoughtful guidance and indispensable support in preparing the illustrative figures. We thank Wanqian Yang for his helpful suggestions and careful editing of the paper. We also thank Leon A. Gatys, Eran Halperin, Andrew C. Miller, Charles Peyser, Pranav Rajpurkar, Ardavan Saeedi, and Jagadish Venkataraman for engaging in valuable discussions throughout this work. This work was partly supported by the NIH/NHLBI Award R01HL148248, NSF Award 1922658 NRT-HDR: FUTURE Foundations, Translation, and Responsibility for Data Science, NSF CAREER Award 2145542, NSF Award 2404476, ONR N00014-23-1-2634, Apple, and Optum.

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

# A  Pairwise independence in binary XOR experiment

In this section, we show that the three variables in the XOR experiment in Section 2.2 are pairwise independent.

Let

$$\mathbf{a}, \mathbf{b} \sim \text{Bernoulli}(0.5)$$
$$\mathbf{c} = \mathbf{a} \text{ XOR } \mathbf{b}.$$

First, we will show that $\mathbf{c} \sim \text{Bernoulli}(0.5)$:

$$
\begin{aligned}
P(\mathbf{c} = 1) &= \sum_{\mathbf{a},\mathbf{b}} P(\mathbf{c} = 1 \mid \mathbf{a} = a, \mathbf{b} = b) P(\mathbf{a} = a) P(\mathbf{b} = b) \\
&= 0.25 \cdot \sum_{\mathbf{a},\mathbf{b}} P(\mathbf{c} = 1 \mid \mathbf{a} = a, \mathbf{b} = b) \\
&= 0.25 \cdot \big[ P(\mathbf{c} = 1 \mid \mathbf{a} = 0, \mathbf{b} = 0) + P(\mathbf{c} = 1 \mid \mathbf{a} = 0, B = 1) \\
&\qquad\qquad + P(\mathbf{c} = 1 \mid \mathbf{a} = 1, \mathbf{b} = 0) + P(\mathbf{c} = 1 \mid \mathbf{a} = 1, \mathbf{b} = 1) \big] \\
&= 0.25 \cdot \big[ 0 + 1 + 1 + 0 \big] \\
&= 0.5.
\end{aligned}
$$

Next, we will show that $\mathbf{c} \mid \mathbf{a} \sim \text{Bernoulli}(0.5)$:

$$P(\mathbf{c} = 1 \mid \mathbf{a}) = \frac{P(\mathbf{a} \mid \mathbf{c} = 1) P(\mathbf{c} = 1)}{P(\mathbf{a})} = 0.5.$$

By symmetry, since $\mathbf{c} \mid \mathbf{a} \sim \text{Bernoulli}(0.5)$, then $\mathbf{c} \mid \mathbf{b} \sim \text{Bernoulli}(0.5)$.

# B  Total correlation lower bound

Our goal in this section is to derive a lower bound on $\mathbf{TC}(\mathbf{m}_1, \ldots, \mathbf{m}_M)$.

We start by describing in Appendix B.1 the sampling procedure for a batch of $(\mathbf{m}_1, \ldots, \mathbf{m}_M)$ tuples. In Appendix B.2, we derive the desired lower bound in Theorem 3.1 (our proof was inspired by Poole et al. [39]'s derivation of the InfoNCE lower bound, which does not rely on an approximation used by Oord et al. [38]). In Appendix B.3, we show that the bound is closed at optimality. Finally, we use the lower bound to define the Symile objective in Appendix B.4.

## B.1  Sampling procedure

We start by describing the sampling procedure for the batch of $N$ $M$-tuples. In contrastive learning, the objective is to differentiate between positive and negative samples constructed from a given batch of matched data. In order to construct these samples, each modality is treated as the anchor in turn, and then for each anchor modality a corresponding set of positive and negative samples is generated.

Let $\gamma$ be arbitrary in $\{1, \ldots, M\}$, let $\mathbf{m}_\gamma$ denote the anchor modality, and let $\mathbf{m}_{-\gamma}$ denote the $M-1$ non-anchor modalities. Let

$$\mathbf{i} \sim \text{Uniform}(\{1, \ldots, N\}) \tag{8}$$

denote the index of the positive $M$-tuple in the batch.

We draw $\mathbf{m}_\gamma$ from $p(\mathbf{m}_\gamma)$ and $\mathbf{m}_{-\gamma,i}$ from $p_{\mathbf{m}_{-\gamma} \mid \mathbf{m}_\gamma}(\mathbf{m}_{-\gamma,i} \mid \mathbf{m}_\gamma)$. We call $(\mathbf{m}_\gamma, \mathbf{m}_{-\gamma,i})$ our *positive* tuple.

For each non-anchor modality $\mathbf{m}_{\ell \neq \gamma}$, we draw $N-1$ samples of $\mathbf{m}_{\ell,j}$ from $p_{\mathbf{m}_\ell}(\mathbf{m}_{\ell,j})$, so that there are $N-1$ total *negative* tuples $(\mathbf{m}_\gamma, \mathbf{m}_{-\gamma,j})$.

Let $\mathbf{M}_{-\gamma} = \{\mathbf{m}_{-\gamma,n}\}_{n=1}^N$ be the set of all samples of non-anchor modalities $\mathbf{m}_{-\gamma}$ in the batch.

This sampling procedure describes the following distribution:

$$p(\mathbf{m}_\gamma, \mathbf{M}_{-\gamma} \mid \mathbf{i} = i) = p(\mathbf{m}_\gamma) \overbrace{p_{\mathbf{m}_{-\gamma} \mid \mathbf{m}_\gamma}(\mathbf{m}_{-\gamma,i} \mid \mathbf{m}_\gamma)}^{\substack{\mathbf{m}_{-\gamma} \text{ from} \\ \text{positive sample}}} \overbrace{\left[ \prod_{\ell \neq \gamma} \prod_{j \neq i} p_{\mathbf{m}_\ell}(\mathbf{m}_{\ell,j}) \right]}^{\substack{\mathbf{m}_{-\gamma} \text{ from} \\ \text{negative samples}}}. \tag{9}$$

Letting $\mathbf{M}_{\ell \neq \gamma} = \{\mathbf{m}_{\ell,n}\}_{n=1}^N$ be the set of all samples of modality $\mathbf{m}_\ell$ in the batch, the following properties hold by Lemma C.1:

$$p(\mathbf{m}_\gamma \mid \mathbf{i} = i) = p(\mathbf{m}_\gamma)$$

$$p(\mathbf{M}_{\ell \neq \gamma} \mid \mathbf{i} = i) = \prod_{j=1}^N p_{\mathbf{m}_\ell}(\mathbf{m}_{\ell,j}) = p(\mathbf{M}_\ell).$$

## B.2  Lower bound on total correlation

We now derive a lower bound on $\mathbf{TC}(\mathbf{m}_1, \ldots, \mathbf{m}_M)$, which we express using the following notation for convenience:

$$\mathbf{TC}(\mathbf{m}_1, \ldots, \mathbf{m}_M) = \mathbf{TC}(\mathbf{m}_\gamma, \{\mathbf{m}_\ell\}_{\ell \neq \gamma}) = D_{\text{KL}}\big(p(\mathbf{m}_\gamma, \mathbf{m}_{-\gamma}) \,\|\, p(\mathbf{m}_\gamma) \prod_{\ell \neq \gamma} p(\mathbf{m}_\ell)\big).$$

**Theorem B.1** (Total Correlation Lower Bound). *Given the distributions in Equations* (8) *and* (9), *for any value $i$ of $\mathbf{i}$ and any scoring function $g$, a multi-sample contrastive lower bound on total correlation is*

$$\mathbf{TC}(\mathbf{m}_\gamma, \{\mathbf{m}_\ell\}_{\ell \neq \gamma}) \geq \log N + \mathop{\mathbb{E}}_{p(\mathbf{m}_\gamma, \mathbf{M}_{-\gamma} \mid \mathbf{i} = i)} \left[ \log \frac{\exp g(\mathbf{m}_\gamma, \mathbf{m}_{-\gamma,i})}{\sum_{j=1}^N \exp g(\mathbf{m}_\gamma, \mathbf{m}_{-\gamma,j})} \right].$$

*Proof.* By Lemmas C.1 and D.1, we have

$$\mathbf{TC}(\mathbf{m}_\gamma, \{\mathbf{m}_\ell\}_{\ell \neq \gamma}) = \mathbf{TC}(\mathbf{m}_\gamma, \{\mathbf{M}_\ell\}_{\ell \neq \gamma} \mid \mathbf{i} = i) \qquad \text{by Lemma D.1}$$

$$= D_{\mathrm{KL}}\big(p(\mathbf{m}_\gamma, \mathbf{M}_{-\gamma} \mid \mathbf{i} = i) \,\|\, p(\mathbf{m}_\gamma \mid \mathbf{i} = i) \prod_{\ell \neq \gamma} p(\mathbf{M}_\ell \mid \mathbf{i} = i)\big)$$

$$= D_{\mathrm{KL}}\big(p(\mathbf{m}_\gamma, \mathbf{M}_{-\gamma} \mid \mathbf{i} = i) \,\|\, p(\mathbf{m}_\gamma) \prod_{\ell \neq \gamma} p(\mathbf{M}_\ell)\big) \qquad \text{by Lemma C.1}$$

$$= \mathop{\mathbb{E}}_{p(\mathbf{m}_\gamma, \mathbf{M}_{-\gamma} \mid \mathbf{i}=i)} \log \overbrace{\frac{p(\mathbf{M}_{-\gamma} \mid \mathbf{m}_\gamma, \mathbf{i} = i)}{\prod_{\ell \neq \gamma} p(\mathbf{M}_\ell)}}^{\substack{\text{total correlation (TC)} \\ \text{likelihood ratio}}}.$$

We call the above likelihood ratio in blue the total correlation (TC) likelihood ratio. We introduce a variational approximation $q(\mathbf{M}_{-\gamma} \mid \mathbf{m}_\gamma, \mathbf{i} = i)$ that has the same support as $p(\mathbf{M}_{-\gamma} \mid \mathbf{m}_\gamma, \mathbf{i} = i)$:

$$\mathbf{TC}(\mathbf{m}_\gamma, \{\mathbf{m}_\ell\}_{\ell \neq \gamma}) = \mathop{\mathbb{E}}_{p(\mathbf{m}_\gamma, \mathbf{M}_{-\gamma} \mid \mathbf{i}=i)} \left[ \log \frac{p(\mathbf{M}_{-\gamma} \mid \mathbf{m}_\gamma, \mathbf{i} = i)}{q(\mathbf{M}_{-\gamma} \mid \mathbf{m}_\gamma, \mathbf{i} = i)} \cdot \frac{q(\mathbf{M}_{-\gamma} \mid \mathbf{m}_\gamma, \mathbf{i} = i)}{\prod_{\ell \neq \gamma} p(\mathbf{M}_\ell)} \right]$$

$$= \mathop{\mathbb{E}}_{p(\mathbf{m}_\gamma, \mathbf{M}_{-\gamma} \mid \mathbf{i}=i)} \left[ \log \frac{p(\mathbf{M}_{-\gamma} \mid \mathbf{m}_\gamma, \mathbf{i} = i)}{q(\mathbf{M}_{-\gamma} \mid \mathbf{m}_\gamma, \mathbf{i} = i)} \right] + \mathop{\mathbb{E}}_{p(\mathbf{m}_\gamma, \mathbf{M}_{-\gamma} \mid \mathbf{i}=i)} \left[ \log \frac{q(\mathbf{M}_{-\gamma} \mid \mathbf{m}_\gamma, \mathbf{i} = i)}{\prod_{\ell \neq \gamma} p(\mathbf{M}_\ell)} \right]$$

$$= \mathop{\mathbb{E}}_{p(\mathbf{m}_\gamma \mid \mathbf{i}=i)} \left[ D_{\mathrm{KL}}\big(p(\mathbf{M}_{-\gamma} \mid \mathbf{m}_\gamma, \mathbf{i} = i) \,\|\, q(\mathbf{M}_{-\gamma} \mid \mathbf{m}_\gamma, \mathbf{i} = i)\big) \right]$$

$$+ \mathop{\mathbb{E}}_{p(\mathbf{m}_\gamma, \mathbf{M}_{-\gamma} \mid \mathbf{i}=i)} \left[ \log \frac{q(\mathbf{M}_{-\gamma} \mid \mathbf{m}_\gamma, \mathbf{i} = i)}{\prod_{\ell \neq \gamma} p(\mathbf{M}_\ell)} \right]$$

$$\geq \mathop{\mathbb{E}}_{p(\mathbf{m}_\gamma, \mathbf{M}_{-\gamma} \mid \mathbf{i}=i)} \left[ \log \frac{q(\mathbf{M}_{-\gamma} \mid \mathbf{m}_\gamma, \mathbf{i} = i)}{\prod_{\ell \neq \gamma} p(\mathbf{M}_\ell)} \right], \tag{10}$$

since the Kullback–Leibler divergence is always non-negative. Note that Equation (10) is the total correlation variant of Barber & Agakov [7]'s lower bound on mutual information.

We choose to set

$$q(\mathbf{M}_{-\gamma} \mid \mathbf{m}_\gamma, \mathbf{i} = i) = \frac{\prod_{\ell \neq \gamma} p(\mathbf{M}_\ell)}{C(\mathbf{m}_\gamma, i)} \exp f(i, \mathbf{m}_\gamma, \mathbf{M}_{-\gamma}) \tag{11}$$

where

$$C(\mathbf{m}_\gamma, i) = \mathop{\mathbb{E}}_{\prod_{\ell \neq \gamma} p(\mathbf{M}_\ell)} \exp f(i, \mathbf{m}_\gamma, \mathbf{M}_{-\gamma})$$

is a normalizing constant,

$$f(i, \mathbf{m}_\gamma, \mathbf{M}_{-\gamma}) = 1 + \log \frac{\exp g(\mathbf{m}_\gamma, \mathbf{m}_{-\gamma, i})}{\frac{1}{N} \sum_{j=1}^N \exp g(\mathbf{m}_\gamma, \mathbf{m}_{-\gamma, j})}, \tag{12}$$

and $g$ is an arbitrary function.

Plugging Equation (11) into Equation (10) gives

$$\mathbf{TC}(\mathbf{m}_\gamma, \{\mathbf{m}_\ell\}_{\ell \neq \gamma}) \geq \mathop{\mathbb{E}}_{p(\mathbf{m}_\gamma, \mathbf{M}_{-\gamma} \mid i=i)} \left[ \log \frac{\frac{\prod_{\ell \neq \gamma} p(\mathbf{M}_\ell)}{C(\mathbf{m}_\gamma, i)} \exp f(i, \mathbf{m}_\gamma, \mathbf{M}_{-\gamma})}{\prod_{\ell \neq \gamma} p(\mathbf{M}_\ell)} \right]$$

$$= \mathop{\mathbb{E}}_{p(\mathbf{m}_\gamma, \mathbf{M}_{-\gamma} \mid i=i)} \left[ f(i, \mathbf{m}_\gamma, \mathbf{M}_{-\gamma}) \right] - \mathop{\mathbb{E}}_{p(\mathbf{m}_\gamma)} \left[ \log C(\mathbf{m}_\gamma, i) \right]. \tag{13}$$

Since $\log(b) \leq \frac{b}{a} + \log a - 1$ for all $b, a > 0$, we see that

$$\log C(\mathbf{m}_\gamma, i) \leq \frac{C(\mathbf{m}_\gamma, i)}{e} + \log e - 1 = \frac{1}{e} C(\mathbf{m}_\gamma, i),$$

which, continuing from Equation (13), gives us

$$\mathbf{TC}(\mathbf{m}_\gamma, \{\mathbf{m}_\ell\}_{\ell \neq \gamma}) \geq \mathop{\mathbb{E}}_{p(\mathbf{m}_\gamma, \mathbf{M}_{-\gamma} \mid i=i)} \left[ f(i, \mathbf{m}_\gamma, \mathbf{M}_{-\gamma}) \right] - \frac{1}{e} \mathop{\mathbb{E}}_{p(\mathbf{m}_\gamma)} \left[ C(\mathbf{m}_\gamma, i) \right]. \tag{14}$$

Substituting the formulas for $f$ and $C$ into Equation (14),

$$\mathbf{TC}(\mathbf{m}_\gamma, \{\mathbf{m}_\ell\}_{\ell\neq\gamma})$$

$$\geq \mathop{\mathbb{E}}_{p(\mathbf{m}_\gamma, \mathbf{M}_{-\gamma} \mid \mathbf{i}=i)}\left[1 + \log \frac{\exp g(\mathbf{m}_\gamma, \mathbf{m}_{-\gamma,i})}{\frac{1}{N}\sum_{j=1}^N \exp g(\mathbf{m}_\gamma, \mathbf{m}_{-\gamma,j})}\right] - \frac{1}{e}\mathop{\mathbb{E}}_{p(\mathbf{m}_\gamma)}\left[\mathop{\mathbb{E}}_{\prod_{\ell\neq\gamma} p(\mathbf{M}_\ell)} \exp\left(1 + \log \frac{\exp g(\mathbf{m}_\gamma, \mathbf{m}_{-\gamma,i})}{\frac{1}{N}\sum_{j=1}^N \exp g(\mathbf{m}_\gamma, \mathbf{m}_{-\gamma,j})}\right)\right]$$

$$= 1 + \mathop{\mathbb{E}}_{p(\mathbf{m}_\gamma, \mathbf{M}_{-\gamma} \mid \mathbf{i}=i)}\left[\log \frac{\exp g(\mathbf{m}_\gamma, \mathbf{m}_{-\gamma,i})}{\frac{1}{N}\sum_{j=1}^N \exp g(\mathbf{m}_\gamma, \mathbf{m}_{-\gamma,j})}\right] - \mathop{\mathbb{E}}_{p(\mathbf{m}_\gamma)\prod_{\ell\neq\gamma} p(\mathbf{M}_\ell)}\left[\frac{\exp g(\mathbf{m}_\gamma, \mathbf{m}_{-\gamma,i})}{\frac{1}{N}\sum_{j=1}^N \exp g(\mathbf{m}_\gamma, \mathbf{m}_{-\gamma,j})}\right].$$

Now take the expectation of this bound over $p(\mathbf{i})$:

$$\mathop{\mathbb{E}}_{p(\mathbf{i})}\left[\mathbf{TC}(\mathbf{m}_\gamma, \{\mathbf{m}_\ell\}_{\ell\neq\gamma})\right]$$

$$\geq \mathop{\mathbb{E}}_{p(\mathbf{i})}\left[1 + \mathop{\mathbb{E}}_{p(\mathbf{m}_\gamma, \mathbf{M}_{-\gamma} \mid \mathbf{i}=i)}\left[\log \frac{\exp g(\mathbf{m}_\gamma, \mathbf{m}_{-\gamma,i})}{\frac{1}{N}\sum_{j=1}^N \exp g(\mathbf{m}_\gamma, \mathbf{m}_{-\gamma,j})}\right] - \mathop{\mathbb{E}}_{p(\mathbf{m}_\gamma)\prod_{\ell\neq\gamma} p(\mathbf{M}_\ell)}\left[\frac{\exp g(\mathbf{m}_\gamma, \mathbf{m}_{-\gamma,i})}{\frac{1}{N}\sum_{j=1}^N \exp g(\mathbf{m}_\gamma, \mathbf{m}_{-\gamma,j})}\right]\right]$$

$$\Longleftrightarrow$$

$$\mathbf{TC}(\mathbf{m}_\gamma, \{\mathbf{m}_\ell\}_{\ell\neq\gamma})$$

$$\geq 1 + \mathop{\mathbb{E}}_{p(\mathbf{m}_\gamma, \mathbf{M}_{-\gamma}, \mathbf{i})}\left[\log \frac{\exp g(\mathbf{m}_\gamma, \mathbf{m}_{-\gamma,\mathbf{i}})}{\frac{1}{N}\sum_{j=1}^N \exp\left[g(\mathbf{m}_\gamma, \mathbf{m}_{-\gamma,j})\right]}\right] - \mathop{\mathbb{E}}_{p(\mathbf{i})p(\mathbf{m}_\gamma)\prod_{\ell\neq\gamma} p(\mathbf{M}_\ell)}\left[\frac{\exp g(\mathbf{m}_\gamma, \mathbf{m}_{-\gamma,\mathbf{i}})}{\frac{1}{N}\sum_{j=1}^N \exp g(\mathbf{m}_\gamma, \mathbf{m}_{-\gamma,j})}\right]$$

$$= 1 + \mathop{\mathbb{E}}_{p(\mathbf{m}_\gamma, \mathbf{M}_{-\gamma}, \mathbf{i})}\left[\log \frac{\exp g(\mathbf{m}_\gamma, \mathbf{m}_{-\gamma,\mathbf{i}})}{\frac{1}{N}\sum_{j=1}^N \exp g(\mathbf{m}_\gamma, \mathbf{m}_{-\gamma,j})}\right] - \overbrace{\mathop{\mathbb{E}}_{p(\mathbf{m}_\gamma)\prod_{\ell\neq\gamma} p(\mathbf{M}_\ell)}\left[\frac{\frac{1}{N}\sum_{i=1}^N \exp g(\mathbf{m}_\gamma, \mathbf{m}_{-\gamma,i})}{\frac{1}{N}\sum_{j=1}^N \exp g(\mathbf{m}_\gamma, \mathbf{m}_{-\gamma,j})}\right]}^{=1}$$

$$= \mathop{\mathbb{E}}_{p(\mathbf{m}_\gamma, \mathbf{M}_{-\gamma}, \mathbf{i})}\left[\log \frac{\exp g(\mathbf{m}_\gamma, \mathbf{m}_{-\gamma,\mathbf{i}})}{\frac{1}{N}\sum_{j=1}^N \exp g(\mathbf{m}_\gamma, \mathbf{m}_{-\gamma,j})}\right]$$

$$= \log N + \mathop{\mathbb{E}}_{p(\mathbf{m}_\gamma, \mathbf{M}_{-\gamma}, \mathbf{i})}\left[\log \frac{\exp g(\mathbf{m}_\gamma, \mathbf{m}_{-\gamma,\mathbf{i}})}{\sum_{j=1}^N \exp g(\mathbf{m}_\gamma, \mathbf{m}_{-\gamma,j})}\right]$$

$$= \log N + \frac{1}{N}\sum_{i=1}^N \mathop{\mathbb{E}}_{p(\mathbf{m}_\gamma, \mathbf{M}_{-\gamma} \mid \mathbf{i}=i)}\left[\log \frac{\exp g(\mathbf{m}_\gamma, \mathbf{m}_{-\gamma,i})}{\sum_{j=1}^N \exp g(\mathbf{m}_\gamma, \mathbf{m}_{-\gamma,j})}\right]. \tag{15}$$

Notice that the index $i$ does not change the expected value in Equation (15). To see why, consider two values $i$ and $i'$:

$$\mathop{\mathbb{E}}_{p(\mathbf{m}_\gamma, \mathbf{M}_{-\gamma} \mid \mathbf{i}=i)}\left[\log \frac{\exp g(\mathbf{m}_\gamma, \mathbf{m}_{-\gamma,i})}{\sum_{j=1}^N \exp g(\mathbf{m}_\gamma, \mathbf{m}_{-\gamma,j})}\right]$$

$$= \int p(\mathbf{m}_\gamma) p_{\mathbf{m}_{-\gamma} \mid \mathbf{m}_\gamma}(\mathbf{m}_{-\gamma,i} \mid \mathbf{m}_\gamma)\left[\prod_{\ell\neq\gamma}\prod_{j\neq i} p_{\mathbf{m}_\ell}(\mathbf{m}_{\ell,j})\right]\left[\log \frac{\exp g(\mathbf{m}_\gamma, \mathbf{m}_{-\gamma,i})}{\sum_{j=1}^N \exp g(\mathbf{m}_\gamma, \mathbf{m}_{-\gamma,j})}\right] d\mathbf{m}_\gamma \, d\mathbf{M}_{-\gamma} \tag{16}$$

$$= \int p(\mathbf{m}_\gamma) p_{\mathbf{m}_{-\gamma} \mid \mathbf{m}_\gamma}(\mathbf{m}_{-\gamma,i'} \mid \mathbf{m}_\gamma)\left[\prod_{\ell\neq\gamma}\prod_{j\neq i'} p_{\mathbf{m}_\ell}(\mathbf{m}_{\ell,j})\right]\left[\log \frac{\exp g(\mathbf{m}_\gamma, \mathbf{m}_{-\gamma,i'})}{\sum_{j=1}^N \exp g(\mathbf{m}_\gamma, \mathbf{m}_{-\gamma,j})}\right] d\mathbf{m}_\gamma \, d\mathbf{M}_{-\gamma} \tag{17}$$

$$= \mathop{\mathbb{E}}_{p(\mathbf{m}_\gamma, \mathbf{M}_{-\gamma} \mid \mathbf{i}=i')}\left[\log \frac{\exp g(\mathbf{m}_\gamma, \mathbf{m}_{-\gamma,i'})}{\sum_{j=1}^N \exp g(\mathbf{m}_\gamma, \mathbf{m}_{-\gamma,j})}\right].$$

Swapping the names of integration variables does not change the integral from Equation (16) to Equation (17).

Therefore, continuing from Equation (15), the lower bound can be written for any value $i$ of $\mathbf{i}$ as

$$\mathbf{TC}(\mathbf{m}_\gamma, \{\mathbf{m}_\ell\}_{\ell\neq\gamma}) \geq \log N + \frac{1}{N}\sum_{i=1}^N \mathop{\mathbb{E}}_{p(\mathbf{m}_\gamma, \mathbf{M}_{-\gamma} \mid \mathbf{i}=i)}\left[\log \frac{\exp g(\mathbf{m}_\gamma, \mathbf{m}_{-\gamma,i})}{\sum_{j=1}^N \exp g(\mathbf{m}_\gamma, \mathbf{m}_{-\gamma,j})}\right]$$

$$= \log N + \mathop{\mathbb{E}}_{p(\mathbf{m}_\gamma, \mathbf{M}_{-\gamma} \mid \mathbf{i}=i)}\left[\log \frac{\exp g(\mathbf{m}_\gamma, \mathbf{m}_{-\gamma,i})}{\sum_{j=1}^N \exp g(\mathbf{m}_\gamma, \mathbf{m}_{-\gamma,j})}\right].$$

$\square$

The extra negative samples are auxiliary random variables for computation in that these random variables do not appear in the target total correlation. This is analogous to the auxiliary random variables used in approximating posteriors and probabilistic modeling [37, 42, 48].

## B.3 Closing the lower bound

There are two inequalities in the derivation for the total correlation lower bound in Theorem B.1: the Barber & Agakov gap in Equation (10) and the log ratio gap in Equation (14). In this section, we show that each of these bounds is closed at optimality.

The Barber & Agakov gap in Equation (10) is closed when

$$D_{\mathrm{KL}}\big(p(\mathbf{M}_{-\gamma} \,|\, \mathbf{m}_\gamma, \mathbf{i} = i) \,\|\, q(\mathbf{M}_{-\gamma} \,|\, \mathbf{m}_\gamma, \mathbf{i} = i)\big) = 0.$$

Therefore, closing the Barber & Agakov gap requires

$$p(\mathbf{M}_{-\gamma} \,|\, \mathbf{m}_\gamma, \mathbf{i} = i) = q(\mathbf{M}_{-\gamma} \,|\, \mathbf{m}_\gamma, \mathbf{i} = i)$$
$$= \frac{\prod_{\ell \neq \gamma} p(\mathbf{M}_\ell)}{C(\mathbf{m}_\gamma, i)} \exp f(i, \mathbf{m}_\gamma, \mathbf{M}_{-\gamma})$$

$$\Longleftrightarrow$$

$$\frac{p(\mathbf{M}_{-\gamma} \,|\, \mathbf{m}_\gamma, \mathbf{i} = i)}{\prod_{\ell \neq \gamma} p(\mathbf{M}_\ell)} = \frac{1}{C(\mathbf{m}_\gamma, i)} \exp f(i, \mathbf{m}_\gamma, \mathbf{M}_{-\gamma})$$

$$\Longleftrightarrow$$

$$\log \frac{p(\mathbf{M}_{-\gamma} \,|\, \mathbf{m}_\gamma, \mathbf{i} = i)}{\prod_{\ell \neq \gamma} p(\mathbf{M}_\ell)} = f(i, \mathbf{m}_\gamma, \mathbf{M}_{-\gamma}) - \log C(\mathbf{m}_\gamma, i)$$

$$\Longleftrightarrow$$

$$f(i, \mathbf{m}_\gamma, \mathbf{M}_{-\gamma}) = \log \frac{p(\mathbf{M}_{-\gamma} \,|\, \mathbf{m}_\gamma, \mathbf{i} = i)}{\prod_{\ell \neq \gamma} p(\mathbf{M}_\ell)} + \log C(\mathbf{m}_\gamma, i). \tag{18}$$

The log ratio gap in Equation (14) is closed when

$$C(\mathbf{m}_\gamma, i) = e.$$

Then by Equation (18), the lower bound is closed if

$$f(i, \mathbf{m}_\gamma, \mathbf{M}_{-\gamma}) = \log \frac{p(\mathbf{M}_{-\gamma} \,|\, \mathbf{m}_\gamma, \mathbf{i} = i)}{\prod_{\ell \neq \gamma} p(\mathbf{M}_\ell)} + 1. \tag{19}$$

By Equation (12),

$$f(i, \mathbf{m}_\gamma, \mathbf{M}_{-\gamma}) = 1 + \log \frac{\exp g(\mathbf{m}_\gamma, \mathbf{m}_{-\gamma,i})}{\frac{1}{N} \sum_{j=1}^N \exp g(\mathbf{m}_\gamma, \mathbf{m}_{-\gamma,j})}$$
$$= \log \frac{p(\mathbf{M}_{-\gamma} \,|\, \mathbf{m}_\gamma, \mathbf{i} = i)}{\prod_{\ell \neq \gamma} p(\mathbf{M}_\ell)} + 1 \qquad \text{by Eq. 19}$$

$$\Longleftrightarrow$$

$$\log \frac{\exp g(\mathbf{m}_\gamma, \mathbf{m}_{-\gamma,i})}{\frac{1}{N} \sum_{j=1}^N \exp g(\mathbf{m}_\gamma, \mathbf{m}_{-\gamma,j})} = \log \frac{p(\mathbf{M}_{-\gamma} \,|\, \mathbf{m}_\gamma, \mathbf{i} = i)}{\prod_{\ell \neq \gamma} p(\mathbf{M}_\ell)}$$

$$\Longleftrightarrow$$

$$\frac{\exp g(\mathbf{m}_\gamma, \mathbf{m}_{-\gamma,i})}{\frac{1}{N} \sum_{j=1}^N \exp g(\mathbf{m}_\gamma, \mathbf{m}_{-\gamma,j})} = \frac{p(\mathbf{M}_{-\gamma} \,|\, \mathbf{m}_\gamma, \mathbf{i} = i)}{\prod_{\ell \neq \gamma} p(\mathbf{M}_\ell)}$$
$$= \frac{p(\mathbf{m}_\gamma, \mathbf{M}_{-\gamma} \,|\, \mathbf{i} = i)}{p(\mathbf{m}_\gamma) \prod_{\ell \neq \gamma} p(\mathbf{M}_\ell)}$$
$$= \frac{p(\mathbf{m}_\gamma) p_{\mathbf{m}_{-\gamma} \,|\, \mathbf{m}_\gamma}(\mathbf{m}_{-\gamma,i} \,|\, \mathbf{m}_\gamma) \Big[ \prod_{\ell \neq \gamma} \prod_{j \neq i} p_{\mathbf{m}_\ell}(\mathbf{m}_{\ell,j}) \Big]}{p(\mathbf{m}_\gamma) \prod_{\ell \neq \gamma} \prod_{k=1}^N p_{\mathbf{m}_\ell}(\mathbf{m}_{\ell,k})} \qquad \text{by Lemma C.1}$$
$$= \frac{p(\mathbf{m}_\gamma) p_{\mathbf{m}_{-\gamma} \,|\, \mathbf{m}_\gamma}(\mathbf{m}_{-\gamma,i} \,|\, \mathbf{m}_\gamma)}{p(\mathbf{m}_\gamma) \prod_{\ell \neq \gamma} p_{\mathbf{m}_\ell}(\mathbf{m}_{\ell,i})}.$$

Therefore, we need

$$\frac{\exp g(\mathbf{m}_\gamma, \mathbf{m}_{-\gamma,i})}{\frac{1}{N} \sum_{j=1}^N \exp g(\mathbf{m}_\gamma, \mathbf{m}_{-\gamma,j})} = \frac{p_{\mathbf{m}_\gamma, \mathbf{m}_{-\gamma}}(\mathbf{m}_\gamma, \mathbf{m}_{-\gamma,i})}{p(\mathbf{m}_\gamma) \prod_{\ell \neq \gamma} p_{\mathbf{m}_\ell}(\mathbf{m}_{\ell,i})}. \tag{20}$$

Let

$$g(\mathbf{m}_\gamma, \mathbf{m}_{-\gamma,i}) = \log \left[ \kappa \frac{p_{\mathbf{m}_\gamma, \mathbf{m}_{-\gamma}}(\mathbf{m}_\gamma, \mathbf{m}_{-\gamma,i})}{p(\mathbf{m}_\gamma) \prod_{\ell \neq \gamma} p_{\mathbf{m}_\ell}(\mathbf{m}_{\ell,i})} \right] \tag{21}$$

$$\Longleftrightarrow$$

$$\exp g(\mathbf{m}_\gamma, \mathbf{m}_{-\gamma,i}) = \kappa \frac{p_{\mathbf{m}_\gamma, \mathbf{m}_{-\gamma}}(\mathbf{m}_\gamma, \mathbf{m}_{-\gamma,i})}{p(\mathbf{m}_\gamma) \prod_{\ell \neq \gamma} p_{\mathbf{m}_\ell}(\mathbf{m}_{\ell,i})}$$

$$\Longleftrightarrow$$

$$\underset{p(\mathbf{m}_\gamma) \prod_{\ell \neq \gamma} p_{\mathbf{m}_\ell}(\mathbf{m}_{\ell,i})}{\mathbb{E}} \exp g(\mathbf{m}_\gamma, \mathbf{m}_{-\gamma,i}) = \underset{p(\mathbf{m}_\gamma) \prod_{\ell \neq \gamma} p_{\mathbf{m}_\ell}(\mathbf{m}_{\ell,i})}{\mathbb{E}} \kappa \frac{p_{\mathbf{m}_\gamma, \mathbf{m}_{-\gamma}}(\mathbf{m}_\gamma, \mathbf{m}_{-\gamma,i})}{p(\mathbf{m}_\gamma) \prod_{\ell \neq \gamma} p_{\mathbf{m}_\ell}(\mathbf{m}_{\ell,i})}$$

$$= \kappa. \tag{22}$$

Informally, for large enough $N$,

$$\frac{1}{N} \sum_{j=1}^N \exp g(\mathbf{m}_\gamma, \mathbf{m}_{-\gamma,j}) \approx \underset{p(\mathbf{m}_\gamma) \prod_{\ell \neq \gamma} p_{\mathbf{m}_\ell}(\mathbf{m}_{\ell,i})}{\mathbb{E}} \exp g(\mathbf{m}_\gamma, \mathbf{m}_{-\gamma,i}).$$

Therefore, we have

$$\frac{\exp g(\mathbf{m}_\gamma, \mathbf{m}_{-\gamma,i})}{\frac{1}{N} \sum_{j=1}^N \exp g(\mathbf{m}_\gamma, \mathbf{m}_{-\gamma,j})} \approx \frac{\exp g(\mathbf{m}_\gamma, \mathbf{m}_{-\gamma,i})}{\mathbb{E}_{p(\mathbf{m}_\gamma) \prod_{\ell \neq \gamma} p_{\mathbf{m}_\ell}(\mathbf{m}_{\ell,i})} \exp g(\mathbf{m}_\gamma, \mathbf{m}_{-\gamma,i})}$$

$$= \frac{\exp g(\mathbf{m}_\gamma, \mathbf{m}_{-\gamma,i})}{\kappa} \qquad \text{by Eq. 22}$$

$$= \frac{1}{\kappa} \exp \log \left[ \kappa \frac{p_{\mathbf{m}_\gamma, \mathbf{m}_{-\gamma}}(\mathbf{m}_\gamma, \mathbf{m}_{-\gamma,i})}{p(\mathbf{m}_\gamma) \prod_{\ell \neq \gamma} p_{\mathbf{m}_\ell}(\mathbf{m}_{\ell,i})} \right] \qquad \text{by Eq. 21}$$

$$= \frac{p_{\mathbf{m}_\gamma, \mathbf{m}_{-\gamma}}(\mathbf{m}_\gamma, \mathbf{m}_{-\gamma,i})}{p(\mathbf{m}_\gamma) \prod_{\ell \neq \gamma} p_{\mathbf{m}_\ell}(\mathbf{m}_{\ell,i})},$$

as required by Equation (20).

The solution for the scoring function $g$ in Equation (21) equals the $g^*$, derived in Lemma E.1, that maximizes the total correlation lower bound.

### B.4 The Symile objective

Given a batch of $N'$ positive tuples $(\mathbf{m}_{\gamma,i}, \mathbf{m}_{-\gamma,i})$, each with $N - 1$ corresponding negative tuples $(\mathbf{m}_{\gamma,i}, \mathbf{m}'_{-\gamma,j})$, and letting $\tau \in \mathbb{R}^+$ be a temperature parameter, the Symile loss is the negative of an empirical estimate of the expected log likelihood in the lower bound in Theorem B.1:

$$\ell^{(\mathbf{m}_\gamma \to \mathbf{m}_{-\gamma})}(\boldsymbol{\theta}, \tau) =$$

$$-\frac{1}{N'} \sum_{i=1}^{N'} \log \frac{\exp\left(\langle f_\gamma^{\boldsymbol{\theta}}(\mathbf{m}_{\gamma,i}), f_{-\gamma}^{\boldsymbol{\theta}}(\mathbf{m}_{-\gamma,i})\rangle / \tau\right)}{\exp\left(\langle f_\gamma^{\boldsymbol{\theta}}(\mathbf{m}_{\gamma,i}), f_{-\gamma}^{\boldsymbol{\theta}}(\mathbf{m}_{-\gamma,i})\rangle / \tau\right) + \sum_{j=1}^{N-1} \exp\left(\langle f_\gamma^{\boldsymbol{\theta}}(\mathbf{m}_{\gamma,i}), f_{-\gamma}^{\boldsymbol{\theta}}(\mathbf{m}'_{-\gamma,j})\rangle / \tau\right)}.$$

## C    Batch sampling procedure properties

**Lemma C.1** (Batch Sampling Procedure Properties). *Suppose a batch of $N$ $M$-tuples is sampled according to the data generating process outlined in Appendix B.1 where*

$$\mathbf{i} \sim Uniform(\{1, \ldots, N\})$$

$$p(\mathbf{m}_\gamma, \mathbf{M}_{-\gamma} \mid \mathbf{i} = i) = p(\mathbf{m}_\gamma) p_{\mathbf{m}_{-\gamma} \mid \mathbf{m}_\gamma}(\mathbf{m}_{-\gamma,i} \mid \mathbf{m}_\gamma) \left[ \prod_{\ell \neq \gamma} \prod_{j \neq i} p_{\mathbf{m}_\ell}(\mathbf{m}_{\ell,j}) \right]. \tag{23}$$

*Let $\mathbf{M}_{\ell \neq \gamma} = \{\mathbf{m}_{\ell,n}\}_{n=1}^N$ be the set of all samples of modality $\mathbf{m}_\ell$ in the batch. The following properties hold:*

$$p(\mathbf{m}_\gamma \mid \mathbf{i} = i) = p(\mathbf{m}_\gamma)$$

$$p(\mathbf{M}_{\ell \neq \gamma} \mid \mathbf{i} = i) = \prod_{j=1}^N p_{\mathbf{m}_\ell}(\mathbf{m}_{\ell,j}) = p(\mathbf{M}_\ell)$$

$$p(\mathbf{m}_\gamma, \mathbf{m}_{-\gamma,i} \mid \mathbf{i} = i) = p(\mathbf{m}_\gamma) p_{\mathbf{m}_{-\gamma} \mid \mathbf{m}_\gamma}(\mathbf{m}_{-\gamma,i} \mid \mathbf{m}_\gamma) = p_{\mathbf{m}_\gamma, \mathbf{m}_{-\gamma}}(\mathbf{m}_\gamma, \mathbf{m}_{-\gamma,i}).$$

*Proof.* **C.1**

**Derive $p(\mathbf{m}_\gamma \mid \mathbf{i} = i) = p(\mathbf{m}_\gamma)$.**

$$
\begin{aligned}
p(\mathbf{m}_\gamma \mid \mathbf{i} = i) &= \int_{\mathbf{M}_{-\gamma}} p(\mathbf{m}_\gamma, \mathbf{M}_{-\gamma} \mid \mathbf{i} = i) \, d\mathbf{M}_{-\gamma} \\
&= \int_{\mathbf{M}_{-\gamma}} p(\mathbf{m}_\gamma) p_{\mathbf{m}_{-\gamma} \mid \mathbf{m}_\gamma}(\mathbf{m}_{-\gamma,i} \mid \mathbf{m}_\gamma) \left[ \prod_{\ell \neq \gamma} \prod_{j \neq i} p_{\mathbf{m}_\ell}(\mathbf{m}_{\ell,j}) \right] d\mathbf{M}_{-\gamma} &\text{by Eq. 23} \\
&= \int_{\mathbf{m}_{-\gamma,i}} p(\mathbf{m}_\gamma) p_{\mathbf{m}_{-\gamma} \mid \mathbf{m}_\gamma}(\mathbf{m}_{-\gamma,i} \mid \mathbf{m}_\gamma) \overbrace{\int_{\mathbf{M}_{-\gamma,j \neq i}} \left[ \prod_{\ell \neq \gamma} \prod_{j \neq i} p_{\mathbf{m}_\ell}(\mathbf{m}_{\ell,j}) \right] d\mathbf{M}_{-\gamma, j \neq i}}^{=1} \, d\mathbf{m}_{-\gamma,i} \\
&= p(\mathbf{m}_\gamma) \overbrace{\int_{\mathbf{m}_{-\gamma,i}} p_{\mathbf{m}_{-\gamma} \mid \mathbf{m}_\gamma}(\mathbf{m}_{-\gamma,i} \mid \mathbf{m}_\gamma) \, d\mathbf{m}_{-\gamma,i}}^{=1} \\
&= p(\mathbf{m}_\gamma).
\end{aligned}
$$

### C.2
**Derive $p(\mathbf{M}_{\ell \neq \gamma} \mid \mathbf{i} = i) = \prod_{j=1}^N p_{\mathbf{m}_\ell}(\mathbf{m}_{\ell,j}) = p(\mathbf{M}_\ell)$.**

Let $\mathbf{M}_{-\gamma} \setminus \mathbf{M}_\ell$ denote the set of all samples of all non-anchor modalities *excluding* the modality $\mathbf{m}_\ell$.

$$
\begin{aligned}
p(\mathbf{M}_{\ell \neq \gamma} \mid \mathbf{i} = i) &= \int_{\mathbf{m}_\gamma} \int_{\mathbf{M}_{-\gamma} \setminus \mathbf{M}_\ell} p(\mathbf{m}_\gamma, \mathbf{M}_{-\gamma} \mid \mathbf{i} = i) \, d\mathbf{M}_{-\gamma} \setminus \mathbf{M}_\ell \, d\mathbf{m}_\gamma \\
&= \int_{\mathbf{m}_\gamma} \int_{\mathbf{M}_{-\gamma} \setminus \mathbf{M}_\ell} p(\mathbf{m}_\gamma) p_{\mathbf{m}_{-\gamma} \mid \mathbf{m}_\gamma}(\mathbf{m}_{-\gamma,i} \mid \mathbf{m}_\gamma) \left[ \prod_{k \neq \gamma} \prod_{j \neq i} p_{\mathbf{m}_k}(\mathbf{m}_{k,j}) \right] d\mathbf{M}_{-\gamma} \setminus \mathbf{M}_\ell \, d\mathbf{m}_\gamma &\text{by Eq. 23} \\
&= \left[ \prod_{j \neq i} p_{\mathbf{m}_\ell}(\mathbf{m}_{\ell,j}) \right] \int_{\mathbf{m}_\gamma} \int_{(\mathbf{m}_{-\gamma} \setminus \mathbf{m}_\ell)_i} p(\mathbf{m}_\gamma) p_{\mathbf{m}_{-\gamma} \mid \mathbf{m}_\gamma}(\mathbf{m}_{-\gamma,i} \mid \mathbf{m}_\gamma) \\
&\qquad \underbrace{\int_{(\mathbf{M}_{-\gamma} \setminus \mathbf{M}_\ell)_{j \neq i}} \left[ \prod_{k \notin \{\gamma, \ell\}} \prod_{j \neq i} p_{\mathbf{m}_k}(\mathbf{m}_{k,j}) \right] d(\mathbf{M}_{-\gamma} \setminus \mathbf{M}_\ell)_{j \neq i} \, d(\mathbf{m}_{-\gamma} \setminus \mathbf{m}_\ell)_i \, d\mathbf{m}_\gamma}_{=1} \\
&= \left[ \prod_{j \neq i} p_{\mathbf{m}_\ell}(\mathbf{m}_{\ell,j}) \right] \int_{\mathbf{m}_\gamma} \int_{(\mathbf{m}_{-\gamma} \setminus \mathbf{m}_\ell)_i} p(\mathbf{m}_\gamma) p_{\mathbf{m}_{-\gamma} \mid \mathbf{m}_\gamma}(\mathbf{m}_{-\gamma,i} \mid \mathbf{m}_\gamma) \, d(\mathbf{m}_{-\gamma} \setminus \mathbf{m}_\ell)_i \, d\mathbf{m}_\gamma \\
&= \left[ \prod_{j \neq i} p_{\mathbf{m}_\ell}(\mathbf{m}_{\ell,j}) \right] p_{\mathbf{m}_\ell}(\mathbf{m}_{\ell,i}) \\
&= \prod_{j=1}^N p_{\mathbf{m}_\ell}(\mathbf{m}_{\ell,j}). \tag{24}
\end{aligned}
$$

We take the expectation over $p(\mathbf{i})$ of both sides of Equation (24) to get

$$\mathop{\mathbb{E}}_{p(\mathbf{i})} p(\mathbf{M}_{\ell \neq \gamma} \mid \mathbf{i} = i) = \mathop{\mathbb{E}}_{p(\mathbf{i})} \prod_{j=1}^{N} p_{\mathbf{m}_\ell}(\mathbf{m}_{\ell,j})$$

$$\Longleftrightarrow$$

$$p(\mathbf{M}_\ell) = \mathop{\mathbb{E}}_{p(\mathbf{i})} \prod_{j=1}^{N} p_{\mathbf{m}_\ell}(\mathbf{m}_{\ell,j})$$

$$= p(\mathbf{M}_{\ell \neq \gamma} \mid \mathbf{i} = i). \qquad\qquad \text{by Eq. 24}$$

## C.3

**Derive** $p(\mathbf{m}_\gamma, \mathbf{m}_{-\gamma,i} \mid \mathbf{i} = i) = p(\mathbf{m}_\gamma) p_{\mathbf{m}_{-\gamma} \mid \mathbf{m}_\gamma}(\mathbf{m}_{-\gamma,i} \mid \mathbf{m}_\gamma) = p_{\mathbf{m}_\gamma, \mathbf{m}_{-\gamma}}(\mathbf{m}_\gamma, \mathbf{m}_{-\gamma,i}).$

$$p(\mathbf{m}_\gamma, \mathbf{m}_{-\gamma,i} \mid \mathbf{i} = i) = \int_{\mathbf{M}_{-\gamma, j \neq i}} p(\mathbf{m}_\gamma, \mathbf{M}_{-\gamma} \mid \mathbf{i} = i) \, d\mathbf{M}_{-\gamma, j \neq i}$$

$$= \int_{\mathbf{M}_{-\gamma, j \neq i}} p(\mathbf{m}_\gamma) p_{\mathbf{m}_{-\gamma} \mid \mathbf{m}_\gamma}(\mathbf{m}_{-\gamma,i} \mid \mathbf{m}_\gamma) \Big[ \prod_{\ell \neq \gamma} \prod_{j \neq i} p_{\mathbf{m}_\ell}(\mathbf{m}_{\ell,j}) \Big] \, d\mathbf{M}_{-\gamma, j \neq i} \quad \text{by Eq. 23}$$

$$= p(\mathbf{m}_\gamma) p_{\mathbf{m}_{-\gamma} \mid \mathbf{m}_\gamma}(\mathbf{m}_{-\gamma,i} \mid \mathbf{m}_\gamma) \overbrace{\int_{\mathbf{M}_{-\gamma, j \neq i}} \prod_{\ell \neq \gamma} \prod_{j \neq i} p_{\mathbf{m}_\ell}(\mathbf{m}_{\ell,j}) \, d\mathbf{M}_{-\gamma, j \neq i}}^{=1}$$

$$= p(\mathbf{m}_\gamma) p_{\mathbf{m}_{-\gamma} \mid \mathbf{m}_\gamma}(\mathbf{m}_{-\gamma,i} \mid \mathbf{m}_\gamma)$$

$$= p_{\mathbf{m}_\gamma, \mathbf{m}_{-\gamma}}(\mathbf{m}_\gamma, \mathbf{m}_{-\gamma,i}).$$

$$\square$$

## D  Total correlation for a batch

**Lemma D.1** (Total Correlation for a Batch of Tuples). *Suppose a batch of $N$ $M$-tuples is sampled according to the data generating process outlined in Appendix B.1 where*

$$\mathbf{i} \sim \text{Uniform}(\{1, \dots, N\})$$

$$p(\mathbf{m}_\gamma, \mathbf{M}_{-\gamma} \,|\, \mathbf{i} = i) = p(\mathbf{m}_\gamma) p_{\mathbf{m}_{-\gamma} \,|\, \mathbf{m}_\gamma}(\mathbf{m}_{-\gamma,i} \,|\, \mathbf{m}_\gamma) \left[ \prod_{\ell \neq \gamma} \prod_{j \neq i} p_{\mathbf{m}_\ell}(\mathbf{m}_{\ell,j}) \right].$$

*We claim that for any value $i$ of $\mathbf{i}$*

$$\mathbf{TC}(\mathbf{m}_\gamma, \{\mathbf{m}_\ell\}_{\ell \neq \gamma}) = \mathbf{TC}(\mathbf{m}_\gamma, \{\mathbf{M}_\ell\}_{\ell \neq \gamma} \,|\, \mathbf{i} = i).$$

*Proof.* By the definition of conditional total correlation,

$$\mathbf{TC}(\mathbf{m}_\gamma, \{\mathbf{M}_\ell\}_{\ell \neq \gamma} \,|\, \mathbf{i} = i) = D_{\text{KL}}\big( p(\mathbf{m}_\gamma, \mathbf{M}_{-\gamma} \,|\, \mathbf{i} = i) \,\|\, p(\mathbf{m}_\gamma \,|\, \mathbf{i} = i) \prod_{\ell \neq \gamma} p(\mathbf{M}_\ell \,|\, \mathbf{i} = i) \big)$$

$$= \mathop{\mathbb{E}}_{p(\mathbf{m}_\gamma, \mathbf{M}_{-\gamma} \,|\, \mathbf{i}=i)} \log \frac{p(\mathbf{m}_\gamma, \mathbf{M}_{-\gamma} \,|\, \mathbf{i} = i)}{p(\mathbf{m}_\gamma \,|\, \mathbf{i} = i) \prod_{\ell \neq \gamma} p(\mathbf{M}_\ell \,|\, \mathbf{i} = i)}$$

$$= \mathop{\mathbb{E}}_{p(\mathbf{m}_\gamma, \mathbf{M}_{-\gamma} \,|\, \mathbf{i}=i)} \log \frac{p(\mathbf{m}_\gamma, \mathbf{M}_{-\gamma} \,|\, \mathbf{i} = i)}{p(\mathbf{m}_\gamma) \prod_{\ell \neq \gamma} p(\mathbf{M}_\ell)} \qquad \text{by Lemma C.1}$$

$$= \mathop{\mathbb{E}}_{p(\mathbf{m}_\gamma, \mathbf{M}_{-\gamma} \,|\, \mathbf{i}=i)} \log \frac{p(\mathbf{m}_\gamma) p_{\mathbf{m}_{-\gamma} \,|\, \mathbf{m}_\gamma}(\mathbf{m}_{-\gamma,i} \,|\, \mathbf{m}_\gamma) \left[ \prod_{\ell \neq \gamma} \prod_{j \neq i} p_{\mathbf{m}_\ell}(\mathbf{m}_{\ell,j}) \right]}{p(\mathbf{m}_\gamma) \left[ \prod_{\ell \neq \gamma} \prod_{k=1}^{N} p_{\mathbf{m}_\ell}(\mathbf{m}_{\ell,k}) \right]} \qquad \text{by Lemma C.1}$$

$$= \mathop{\mathbb{E}}_{p_{\mathbf{m}_\gamma, \mathbf{m}_{-\gamma}}(\mathbf{m}_\gamma, \mathbf{m}_{-\gamma,i})} \log \frac{p(\mathbf{m}_\gamma) p_{\mathbf{m}_{-\gamma} \,|\, \mathbf{m}_\gamma}(\mathbf{m}_{-\gamma,i} \,|\, \mathbf{m}_\gamma)}{p(\mathbf{m}_\gamma) \prod_{\ell \neq \gamma} p_{\mathbf{m}_\ell}(\mathbf{m}_{\ell,i})}$$

$$= D_{\text{KL}}\big( p_{\mathbf{m}_\gamma, \mathbf{m}_{-\gamma}}(\mathbf{m}_\gamma, \mathbf{m}_{-\gamma,i}) \,\|\, p(\mathbf{m}_\gamma) \prod_{\ell \neq \gamma} p_{\mathbf{m}_\ell}(\mathbf{m}_{\ell,i}) \big)$$

$$= \mathbf{TC}(\mathbf{m}_\gamma, \{\mathbf{m}_\ell\}_{\ell \neq \gamma}).$$

$\square$

# E    Scoring function as total correlation likelihood ratio estimator

In this section, we show that the optimal scoring function is equal to the log total correlation likelihood ratio up to additive constants.

**Lemma E.1** (Scoring Function as Total Correlation Likelihood Ratio Estimator). *Suppose a batch of $N$ $M$-tuples is sampled according to the data generating process outlined in Appendix B.1. For some $\kappa > 0$, the $g$ that maximizes the lower bound*

$$\mathbf{TC}(\mathbf{m}_\gamma, \{\mathbf{m}_\ell\}_{\ell \neq \gamma}) \geq \log N + \underset{p(\mathbf{m}_\gamma, \mathbf{M}_{-\gamma} \mid \mathbf{i}=i)}{\mathbb{E}} \left[ \log \frac{\exp g(\mathbf{m}_\gamma, \mathbf{m}_{-\gamma,i})}{\sum_{j=1}^N \exp g(\mathbf{m}_\gamma, \mathbf{m}_{-\gamma,j})} \right]$$

*is*

$$g^*(\mathbf{m}_\gamma, \mathbf{m}_{-\gamma}) = \kappa + \log \left[ \frac{p_{\mathbf{m}_\gamma, \mathbf{m}_{-\gamma}}(\mathbf{m}_\gamma, \mathbf{m}_{-\gamma})}{p(\mathbf{m}_\gamma) \prod_{\ell \neq \gamma} p_{\mathbf{m}_\ell}(\mathbf{m}_\ell)} \right].$$

*Proof.* Define

$$p^g(\mathbf{i} = i \mid \mathbf{m}_\gamma, \mathbf{M}_{-\gamma}) = \frac{\exp g(\mathbf{m}_\gamma, \mathbf{m}_{-\gamma,i})}{\sum_{j=1}^N \exp g(\mathbf{m}_\gamma, \mathbf{m}_{-\gamma,j})}$$

to be the categorical cross-entropy of correctly classifying the positive tuple $(\mathbf{m}_\gamma, \mathbf{m}_{-\gamma,i})$.

The maximizer of the log likelihood is the true conditional distribution, which by Lemma F.1 is

$$p(i = \text{positive} \mid \mathbf{m}_\gamma, \mathbf{M}_{-\gamma}) = \frac{\frac{p_{\mathbf{m}_{-\gamma} \mid \mathbf{m}_\gamma}(\mathbf{m}_{-\gamma,i} \mid \mathbf{m}_\gamma)}{\prod_{\ell \neq \gamma} p_{\mathbf{m}_\ell}(\mathbf{m}_{\ell,i})}}{\sum_{j=1}^N \frac{p_{\mathbf{m}_{-\gamma} \mid \mathbf{m}_\gamma}(\mathbf{m}_{-\gamma,j} \mid \mathbf{m}_\gamma)}{\prod_{\ell \neq \gamma} p_{\mathbf{m}_\ell}(\mathbf{m}_{\ell,j})}}.$$

Therefore, solving for the form of the optimal $g^*$, we have

$$p(i = \text{positive} \mid \mathbf{m}_\gamma, \mathbf{M}_{-\gamma}) = p^{g^*}(\mathbf{i} = i \mid \mathbf{m}_\gamma, \mathbf{M}_{-\gamma})$$
$$\iff$$
$$\frac{\frac{p_{\mathbf{m}_{-\gamma} \mid \mathbf{m}_\gamma}(\mathbf{m}_{-\gamma,i} \mid \mathbf{m}_\gamma)}{\prod_{\ell \neq \gamma} p_{\mathbf{m}_\ell}(\mathbf{m}_{\ell,i})}}{\sum_{j=1}^N \frac{p_{\mathbf{m}_{-\gamma} \mid \mathbf{m}_\gamma}(\mathbf{m}_{-\gamma,j} \mid \mathbf{m}_\gamma)}{\prod_{\ell \neq \gamma} p_{\mathbf{m}_\ell}(\mathbf{m}_{\ell,j})}} = \frac{\exp g^*(\mathbf{m}_\gamma, \mathbf{m}_{-\gamma,i})}{\sum_{j=1}^N \exp g^*(\mathbf{m}_\gamma, \mathbf{m}_{-\gamma,j})}.$$

Therefore, at optimality, when our model is equal to the true conditional distribution, for some constant $\kappa > 0$, we have

$$g^*(\mathbf{m}_\gamma, \mathbf{m}_{-\gamma}) = \kappa + \log \left[ \frac{p_{\mathbf{m}_\gamma, \mathbf{m}_{-\gamma}}(\mathbf{m}_\gamma, \mathbf{m}_{-\gamma})}{p(\mathbf{m}_\gamma) \prod_{\ell \neq \gamma} p_{\mathbf{m}_\ell}(\mathbf{m}_\ell)} \right].$$

$\square$

# F   Ratio of total correlation likelihood ratios

**Lemma F.1** (Ratio of Total Correlation Likelihood Ratios). *Suppose a batch of $N$ $M$-tuples is sampled according to the data generating process outlined in Appendix B.1 where*

$$\mathbf{i} \sim Uniform(\{1, \ldots, N\})$$

$$p(\mathbf{m}_\gamma, \mathbf{M}_{-\gamma} \,|\, \mathbf{i} = i) = p(\mathbf{m}_\gamma) p_{\mathbf{m}_{-\gamma} \,|\, \mathbf{m}_\gamma}(\mathbf{m}_{-\gamma,i} \,|\, \mathbf{m}_\gamma) \left[ \prod_{\ell \neq \gamma} \prod_{j \neq i} p_{\mathbf{m}_\ell}(\mathbf{m}_{\ell,j}) \right]. \tag{25}$$

*The true conditional probability that $(\mathbf{m}_\gamma, \mathbf{m}_{-\gamma,i})$ is the positive tuple among all $N$ samples in the batch can be expressed as a ratio of total correlation likelihood ratios:*

$$p(i = positive \,|\, \mathbf{m}_\gamma, \mathbf{M}_{-\gamma}) = \frac{\frac{p_{\mathbf{m}_{-\gamma} \,|\, \mathbf{m}_\gamma}(\mathbf{m}_{-\gamma,i} \,|\, \mathbf{m}_\gamma)}{\prod_{\ell \neq \gamma} p_{\mathbf{m}_\ell}(\mathbf{m}_{\ell,i})}}{\sum_{j=1}^{N} \frac{p_{\mathbf{m}_{-\gamma} \,|\, \mathbf{m}_\gamma}(\mathbf{m}_{-\gamma,j} \,|\, \mathbf{m}_\gamma)}{\prod_{\ell \neq \gamma} p_{\mathbf{m}_\ell}(\mathbf{m}_{\ell,j})}}.$$

*Proof.* We first apply the definition of conditional probability and the law of total probability:

$$
\begin{aligned}
p(i = \text{positive} \,|\, \mathbf{m}_\gamma, \mathbf{M}_{-\gamma}) &= \frac{p(\mathbf{m}_\gamma, \mathbf{M}_{-\gamma}, i = \text{positive})}{p(\mathbf{m}_\gamma, \mathbf{M}_{-\gamma})} \\
&= \frac{p(\mathbf{m}_\gamma, \mathbf{M}_{-\gamma}, i = \text{positive})}{\sum_{j=1}^{N} p(\mathbf{m}_\gamma, \mathbf{M}_{-\gamma}, j = \text{positive})} \\
&= \frac{p(\mathbf{m}_\gamma) p_{\mathbf{m}_{-\gamma} \,|\, \mathbf{m}_\gamma}(\mathbf{m}_{-\gamma,i} \,|\, \mathbf{m}_\gamma) \left[ \prod_{\ell \neq \gamma} \prod_{k \neq i} p_{\mathbf{m}_\ell}(\mathbf{m}_{\ell,k}) \right]}{\sum_{j=1}^{N} p(\mathbf{m}_\gamma) p_{\mathbf{m}_{-\gamma} \,|\, \mathbf{m}_\gamma}(\mathbf{m}_{-\gamma,j} \,|\, \mathbf{m}_\gamma) \left[ \prod_{\ell \neq \gamma} \prod_{r \neq j} p_{\mathbf{m}_\ell}(\mathbf{m}_{\ell,r}) \right]} \quad \text{by Eq. 25} \\
&= \frac{\overbrace{\frac{\prod_{\ell \neq \gamma} p_{\mathbf{m}_\ell}(\mathbf{m}_{\ell,i})}{\prod_{\ell \neq \gamma} p_{\mathbf{m}_\ell}(\mathbf{m}_{\ell,i})}}^{=1} p_{\mathbf{m}_{-\gamma} \,|\, \mathbf{m}_\gamma}(\mathbf{m}_{-\gamma,i} \,|\, \mathbf{m}_\gamma) \left[ \prod_{\ell \neq \gamma} \prod_{k \neq i} p_{\mathbf{m}_\ell}(\mathbf{m}_{\ell,k}) \right]}{\sum_{j=1}^{N} \underbrace{\frac{\prod_{\ell \neq \gamma} p_{\mathbf{m}_\ell}(\mathbf{m}_{\ell,j})}{\prod_{\ell \neq \gamma} p_{\mathbf{m}_\ell}(\mathbf{m}_{\ell,j})}}_{=1} p_{\mathbf{m}_{-\gamma} \,|\, \mathbf{m}_\gamma}(\mathbf{m}_{-\gamma,j} \,|\, \mathbf{m}_\gamma) \left[ \prod_{\ell \neq \gamma} \prod_{r \neq j} p_{\mathbf{m}_\ell}(\mathbf{m}_{\ell,r}) \right]} \\
&= \frac{\frac{p_{\mathbf{m}_{-\gamma} \,|\, \mathbf{m}_\gamma}(\mathbf{m}_{-\gamma,i} \,|\, \mathbf{m}_\gamma)}{\prod_{\ell \neq \gamma} p_{\mathbf{m}_\ell}(\mathbf{m}_{\ell,i})} \left[ \prod_{\ell \neq \gamma} \prod_{k=1}^{N} p_{\mathbf{m}_\ell}(\mathbf{m}_{\ell,k}) \right]}{\sum_{j=1}^{N} \frac{p_{\mathbf{m}_{-\gamma} \,|\, \mathbf{m}_\gamma}(\mathbf{m}_{-\gamma,j} \,|\, \mathbf{m}_\gamma)}{\prod_{\ell \neq \gamma} p_{\mathbf{m}_\ell}(\mathbf{m}_{\ell,j})} \left[ \prod_{\ell \neq \gamma} \prod_{r=1}^{N} p_{\mathbf{m}_\ell}(\mathbf{m}_{\ell,r}) \right]} \\
&= \frac{\frac{p_{\mathbf{m}_{-\gamma} \,|\, \mathbf{m}_\gamma}(\mathbf{m}_{-\gamma,i} \,|\, \mathbf{m}_\gamma)}{\prod_{\ell \neq \gamma} p_{\mathbf{m}_\ell}(\mathbf{m}_{\ell,i})}}{\sum_{j=1}^{N} \frac{p_{\mathbf{m}_{-\gamma} \,|\, \mathbf{m}_\gamma}(\mathbf{m}_{-\gamma,j} \,|\, \mathbf{m}_\gamma)}{\prod_{\ell \neq \gamma} p_{\mathbf{m}_\ell}(\mathbf{m}_{\ell,j})}}.
\end{aligned}
$$

$\square$

# G   Symile learns sufficient statistics

**Theorem G.1** (Symile Sufficient Statistics). *Let $\mathbf{m}_1, \ldots, \mathbf{m}_M$ be $M$ random variables whose optimal representations when trained using Symile are $f_1^*(\mathbf{m}_1), \ldots, f_M^*(\mathbf{m}_M)$, respectively. The element-wise product of any subset of the representations is a sufficient statistic for predicting the remaining random variables.*

*For example, letting $\gamma$ be arbitrary in $\{1, \ldots, M\}$ and letting $\prod_{k \neq \gamma} f_k^*(\mathbf{m}_k)$ indicate the element-wise product of the representations for the remaining $M - 1$ modalities, $\prod_{k \neq \gamma} f_k^*(\mathbf{m}_k)$ is a sufficient statistic for predicting $\mathbf{m}_\gamma$, which can be expressed using the following conditional independence statement:*

$$\mathbf{m}_\gamma \perp\!\!\!\perp \mathbf{m}_{-\gamma} \mid \prod_{k \neq \gamma} f_k^*(\mathbf{m}_k).$$

*Proof.* Since, as discussed in Section 3.2, we use the multilinear inner product (MIP) as the scoring function $g$, by Lemma E.1 for some $\kappa > 0$ at optimality, we have

$$g^*(\mathbf{m}_\gamma, \mathbf{m}_{-\gamma}) = \langle \{f_i^*(\mathbf{m}_i)\}_{i=1}^M \rangle = \log \left[ \kappa \frac{p_{\mathbf{m}_\gamma, \mathbf{m}_{-\gamma}}(\mathbf{m}_\gamma, \mathbf{m}_{-\gamma})}{p(\mathbf{m}_\gamma) \prod_{k \neq \gamma} p_{\mathbf{m}_k}(\mathbf{m}_k)} \right]. \tag{26}$$

Consider the case in which we are given representations for the $M - 1$ modalities that are not $\mathbf{m}_\gamma$. The goal is to show

$$\mathbf{m}_\gamma \perp\!\!\!\perp \mathbf{m}_{-\gamma} \mid \prod_{k \neq \gamma} f_k^*(\mathbf{m}_k).$$

To do so, we will show that

$$p\big(\mathbf{m}_\gamma \mid \prod_{k \neq \gamma} f_k^*(\mathbf{m}_k)\big) = p\big(\mathbf{m}_\gamma \mid \mathbf{m}_{-\gamma}, \prod_{k \neq \gamma} f_k^*(\mathbf{m}_k)\big).$$

Since, conditioned on $\mathbf{m}_{-\gamma}$, $\mathbf{m}_\gamma$ is independent of any function of $\mathbf{m}_{k \neq \gamma}$,

$$p\big(\mathbf{m}_\gamma \mid \mathbf{m}_{-\gamma}, \prod_{k \neq \gamma} f_k^*(\mathbf{m}_k)\big) = p(\mathbf{m}_\gamma \mid \mathbf{m}_{-\gamma})$$

$$= \frac{p(\mathbf{m}_\gamma, \mathbf{m}_{-\gamma})}{p(\mathbf{m}_{-\gamma})}$$

$$= \frac{p(\mathbf{m}_\gamma, \mathbf{m}_{-\gamma})}{p(\mathbf{m}_{-\gamma})} \cdot \frac{\kappa \prod_{\ell=1}^M p(\mathbf{m}_\ell)}{\kappa \prod_{\ell=1}^M p(\mathbf{m}_\ell)}$$

$$= \frac{\exp\left[ \langle \{f_i^*(\mathbf{m}_i)\}_{i=1}^M \rangle \right] \prod_{\ell=1}^M p(\mathbf{m}_\ell)}{\kappa \cdot p(\mathbf{m}_{-\gamma})} \quad \text{by Eq. 26.} \tag{27}$$

Since $p$ is a distribution,

$$\int_{\mathbf{m}_\gamma} \frac{\exp\left[ \langle \{f_i^*(\mathbf{m}_i)\}_{i=1}^M \rangle \right] \prod_{\ell=1}^M p(\mathbf{m}_\ell)}{\kappa \cdot p(\mathbf{m}_{-\gamma})} d\mathbf{m}_\gamma = 1$$

$$\Longleftrightarrow$$

$$\int_{\mathbf{m}_\gamma} \exp\left[ \langle \{f_i^*(\mathbf{m}_i)\}_{i=1}^M \rangle \right] p(\mathbf{m}_\gamma) d\mathbf{m}_\gamma = \frac{\kappa \cdot p(\mathbf{m}_{-\gamma})}{\prod_{k \neq \gamma} p(\mathbf{m}_k)}.$$

Substituting this back into Equation (27) yields

$$p\big(\mathbf{m}_\gamma \mid \mathbf{m}_{-\gamma}, \prod_{k \neq \gamma} f_k^*(\mathbf{m}_k)\big) = \frac{\exp\left[ \langle \{f_i^*(\mathbf{m}_i)\}_{i=1}^M \rangle \right] p(\mathbf{m}_\gamma)}{\int_{\mathbf{m}_\gamma} \exp\left[ \langle \{f_i^*(\mathbf{m}_i)\}_{i=1}^M \rangle \right] p(\mathbf{m}_\gamma) d\mathbf{m}_\gamma}. \tag{28}$$

Now compute

$$p\big(\mathbf{m}_\gamma \mid \prod_{k \neq \gamma} f_k^*(\mathbf{m}_k)\big) = \mathop{\mathbb{E}}_{p(\mathbf{m}_{-\gamma} \mid \prod_{k \neq \gamma} f_k^*(\mathbf{m}_k))} p\big(\mathbf{m}_\gamma \mid \mathbf{m}_{-\gamma}, \prod_{k \neq \gamma} f_k^*(\mathbf{m}_k)\big)$$

$$= \underset{p(\mathbf{m}_{-\gamma} \mid \prod_{k \neq \gamma} f_k^*(\mathbf{m}_k))}{\mathbb{E}} \left[ \frac{\exp\left[\langle\{f_i^*(\mathbf{m}_i)\}_{i=1}^M\rangle\right]p(\mathbf{m}_\gamma)}{\int_{\mathbf{m}_\gamma} \exp\left[\langle\{f_i^*(\mathbf{m}_i)\}_{i=1}^M\rangle\right]p(\mathbf{m}_\gamma)d\mathbf{m}_\gamma} \right] \qquad \text{by Eq. 28}$$

$$= \underset{p(\mathbf{m}_{-\gamma} \mid \prod_{k \neq \gamma} f_k^*(\mathbf{m}_k))}{\mathbb{E}} \left[ \frac{\exp\left[\left(\prod_{k \neq \gamma} f_k^*(\mathbf{m}_k)\right)^\top f_\gamma^*(\mathbf{m}_\gamma)\right]p(\mathbf{m}_\gamma)}{\int_{\mathbf{m}_\gamma} \exp\left[\left(\prod_{k \neq \gamma} f_k^*(\mathbf{m}_k)\right)^\top f_\gamma^*(\mathbf{m}_\gamma)\right]p(\mathbf{m}_\gamma)d\mathbf{m}_\gamma} \right].$$

Since $\mathbf{m}_{-\gamma}$ only appears inside the expectation through $\prod_{k \neq \gamma} f_k^*(\mathbf{m}_k)$, and since we are conditioning on $\prod_{k \neq \gamma} f_k^*(\mathbf{m}_k)$ being a particular value, the term inside the expectation is conditionally constant. Therefore,

$$p\big(\mathbf{m}_\gamma \mid \prod_{k \neq \gamma} f_k^*(\mathbf{m}_k)\big) = \frac{\exp\left[\left(\prod_{k \neq \gamma} f_k^*(\mathbf{m}_k)\right)^\top f_\gamma^*(\mathbf{m}_\gamma)\right]p(\mathbf{m}_\gamma)}{\int_{\mathbf{m}_\gamma} \exp\left[\left(\prod_{k \neq \gamma} f_k^*(\mathbf{m}_k)\right)^\top f_\gamma^*(\mathbf{m}_\gamma)\right]p(\mathbf{m}_\gamma)d\mathbf{m}_\gamma}$$

$$= \frac{\exp\left[\langle\{f_i^*(\mathbf{m}_i)\}_{i=1}^M\rangle\right]p(\mathbf{m}_\gamma)}{\int_{\mathbf{m}_\gamma} \exp\left[\langle\{f_i^*(\mathbf{m}_i)\}_{i=1}^M\rangle\right]p(\mathbf{m}_\gamma)d\mathbf{m}_\gamma}$$

$$= p\big(\mathbf{m}_\gamma \mid \mathbf{m}_{-\gamma}, \prod_{k \neq \gamma} f_k^*(\mathbf{m}_k)\big). \qquad \text{by Eq. 28}$$

This equality establishes that

$$\mathbf{m}_\gamma \perp\!\!\!\perp \mathbf{m}_{-\gamma} \mid \prod_{k \neq \gamma} f_k^*(\mathbf{m}_k).$$

$\square$

## H  Zero-shot prediction using the score function

In this section, we discuss the limitations—for both Symile and CLIP—of using the scoring function for zero-shot prediction and demonstrate how these limitations can be addressed by using the scoring function to directly compute the desired conditional probability.

Recall from Lemma 3.2 that the optimal scoring function $g^*$ is equal to the instantaneous total correlation up to additive constants:

$$g^*(\mathbf{x}, \mathbf{y}, \mathbf{z}) = \log \left[ \kappa \frac{p_{\mathbf{x},\mathbf{y},\mathbf{z}}(\mathbf{x}, \mathbf{y}, \mathbf{z})}{p(\mathbf{x}) p_{\mathbf{y}}(\mathbf{y}) p_{\mathbf{z}}(\mathbf{z})} \right].$$

Similarly, the optimal scoring function $h^*$ for CLIP can be expressed as follows [38, 39]:

$$h^*(\mathbf{x}, \mathbf{y}) = \log \left[ \kappa \frac{p_{\mathbf{x},\mathbf{y}}(\mathbf{x}, \mathbf{y})}{p(\mathbf{x}) p_{\mathbf{y}}(\mathbf{y})} \right].$$

Traditionally, for zero-shot prediction with CLIP, the scoring function is used to rank the candidates for one of the modalities: $\arg\max_{y \in \mathcal{Y}} p(\mathbf{y} = y \,|\, x) = \arg\max_{y \in \mathcal{Y}} h^*(x, y)$. However, it turns out that this approach for zero-shot prediction does not lead to the Bayes optimal prediction, potentially sacrificing accuracy.

To illustrate the issue, consider a scenario in which we have two modalities: disease $\mathbf{y}$ and temperature $\mathbf{t}$. The values these two variables can take are outlined in the following joint distribution table:

| y \ t | 99 | 100 | 101 | 102 | $p(\mathbf{y})$ |
|---|---|---|---|---|---|
| $a$ | 0.1 | 0.1 | 0.3 | 0.3 | 0.8 |
| $b$ | 0 | 0 | 0.1 | 0.1 | 0.2 |
| $p(\mathbf{t})$ | 0.1 | 0.1 | 0.4 | 0.4 | |

Now, consider a patient with a temperature of 101 degrees; our goal is to predict which disease the patient has. Predictions derived from the conditional distribution achieve optimal accuracy [36]. Therefore, we should predict that the patient has disease $a$, since

$$p(\mathbf{y} = a \,|\, \mathbf{t} = 101) = \frac{p(\mathbf{y} = a, \mathbf{t} = 101)}{p(\mathbf{t} = 101)} = \frac{0.3}{0.4} = 0.75$$

and

$$p(\mathbf{y} = b \,|\, \mathbf{t} = 101) = \frac{p(\mathbf{y} = b, \mathbf{t} = 101)}{p(\mathbf{t} = 101)} = \frac{0.1}{0.4} = 0.25.$$

However, were we to apply the standard strategy of using the scoring function for zero-shot classification, we would predict that the patient has disease $b$, since dividing by the prior probability of disease $b$ upweights its likelihood ratio compared to that of disease $a$:

$$\frac{p(\mathbf{y} = a \,|\, \mathbf{t} = 101)}{p(\mathbf{y} = a)} = \frac{0.75}{0.8} = 0.9375$$

compared to

$$\frac{p(\mathbf{y} = b \,|\, \mathbf{t} = 101)}{p(\mathbf{y} = b)} = \frac{0.25}{0.2} = 1.25.$$

Why, then, does CLIP perform well in practice? Because the kinds of zero-shot classification tasks for which the dot product is used typically feature an almost deterministic likelihood, where the modality to predict has a point mass distribution at a single value, with probability zero everywhere else.

For example, in our case, this would mean that $p(\mathbf{y} = a \,|\, \mathbf{t} = 101) = 1$ and $p(\mathbf{y} = b \,|\, \mathbf{t} = 101) = 0$, resulting—appropriately—in a higher likelihood ratio for disease $a$ compared to disease $b$:

$$\frac{p(\mathbf{y} = a \,|\, \mathbf{t} = 101)}{p(\mathbf{y} = a)} = \frac{1}{0.8} > \frac{0}{0.2} = \frac{p(\mathbf{y} = b \,|\, \mathbf{t} = 101)}{p(\mathbf{y} = b)}.$$

While zero-shot classification works well when one modality directly determines another (for example, a text caption precisely specifies its corresponding image), in all other instances, the CLIP or Symile scoring function fails to provide reliable predictions.

To address this issue, we demonstrate how the Symile scoring function can be used to compute the desired conditional distribution, which achieves optimal classification accuracy. (While we illustrate this approach for Symile, it can be applied similarly to CLIP.)

Suppose we want to predict modality $\mathbf{y}$ from modalities $\mathbf{x}, \mathbf{z}$ using zero-shot classification. Recall from Section 3.2 that we use the multilinear inner product (MIP) as the scoring function. Theorem H.1 establishes that we can compute $p(\mathbf{y} \mid \mathbf{x}, \mathbf{z})$ directly using the MIP.

**Theorem H.1** (Conditional Distribution using the Scoring Function). *Let $\mathbf{x}, \mathbf{y}, \mathbf{z}$ be three random variables whose optimal representations when trained using Symile are $f_{\mathbf{x}}^*(\mathbf{x}), f_{\mathbf{y}}^*(\mathbf{y}), f_{\mathbf{z}}^*(\mathbf{z})$, respectively. Let the* MIP *$\langle f_{\mathbf{x}}^*(\mathbf{x}), f_{\mathbf{y}}^*(\mathbf{y}), f_{\mathbf{z}}^*(\mathbf{z}) \rangle$ be the scoring function. Then,*

$$p(\mathbf{y} \mid \mathbf{x}, \mathbf{z}) = \frac{\exp\left[\langle f_{\mathbf{x}}^*(\mathbf{x}), f_{\mathbf{y}}^*(\mathbf{y}), f_{\mathbf{z}}^*(\mathbf{z})\rangle\right] p(\mathbf{y})}{\int_{\mathbf{y}} \exp\left[\langle f_{\mathbf{x}}^*(\mathbf{x}), f_{\mathbf{y}}^*(\mathbf{y}), f_{\mathbf{z}}^*(\mathbf{z})\rangle\right] p(\mathbf{y}) d\mathbf{y}}. \tag{29}$$

*Proof.* Let $f_{\mathbf{x}}^*(\mathbf{x}) \odot f_{\mathbf{z}}^*(\mathbf{z})$ indicate the element-wise product of the two representations. Since $f_{\mathbf{x}}^*(\mathbf{x}) \odot f_{\mathbf{z}}^*(\mathbf{z})$ is determined by $\mathbf{x}$ and $\mathbf{z}$,

$$\begin{aligned}
p(\mathbf{y} \mid \mathbf{x}, \mathbf{z}) &= p(\mathbf{y} \mid \mathbf{x}, \mathbf{z}, f_{\mathbf{x}}^*(\mathbf{x}) \odot f_{\mathbf{z}}^*(\mathbf{z})) \\
&= \frac{\exp\left[\langle f_{\mathbf{x}}^*(\mathbf{x}), f_{\mathbf{y}}^*(\mathbf{y}), f_{\mathbf{z}}^*(\mathbf{z})\rangle\right] p(\mathbf{y})}{\int_{\mathbf{y}} \exp\left[\langle f_{\mathbf{x}}^*(\mathbf{x}), f_{\mathbf{y}}^*(\mathbf{y}), f_{\mathbf{z}}^*(\mathbf{z})\rangle\right] p(\mathbf{y}) d\mathbf{y}} \quad \text{by Eq. 28.}
\end{aligned}$$

$\square$

If the marginal distribution of $\mathbf{y}$ is known, we could then perform zero-shot classification in one of two ways. When the distribution $p(\mathbf{y} \mid \mathbf{x}, \mathbf{z})$ itself is of interest, as is often the case in healthcare [10], we could compute $p(\mathbf{y} \mid \mathbf{x}, \mathbf{z})$ directly, following Equation (29). Alternatively, if only predictions are needed, we could use

$$\langle f_{\mathbf{x}}^*(\mathbf{x}), f_{\mathbf{y}}^*(\mathbf{y}), f_{\mathbf{z}}^*(\mathbf{z}) \rangle + \log p(\mathbf{y})$$

to rank the possible values for $\mathbf{y}$.

To see why the latter approach works, first notice that

$$\begin{aligned}
\log p(\mathbf{y} \mid \mathbf{x}, \mathbf{z}) &= \log \frac{\exp\left[\langle f_{\mathbf{x}}^*(\mathbf{x}), f_{\mathbf{y}}^*(\mathbf{y}), f_{\mathbf{z}}^*(\mathbf{z})\rangle\right] p(\mathbf{y})}{\int_{\mathbf{y}} \exp\left[\langle f_{\mathbf{x}}^*(\mathbf{x}), f_{\mathbf{y}}^*(\mathbf{y}), f_{\mathbf{z}}^*(\mathbf{z})\rangle\right] p(\mathbf{y}) d\mathbf{y}} \quad \text{by Eq. 29} \\
&= \langle f_{\mathbf{x}}^*(\mathbf{x}), f_{\mathbf{y}}^*(\mathbf{y}), f_{\mathbf{z}}^*(\mathbf{z}) \rangle + \log p(\mathbf{y}) - \int_{\mathbf{y}} \exp\left[\langle f_{\mathbf{x}}^*(\mathbf{x}), f_{\mathbf{y}}^*(\mathbf{y}), f_{\mathbf{z}}^*(\mathbf{z})\rangle\right] p(\mathbf{y}) d\mathbf{y}
\end{aligned}$$

$$\Longleftrightarrow$$

$$\begin{aligned}
\log p(\mathbf{y} \mid \mathbf{x}, \mathbf{z}) + \int_{\mathbf{y}} & \exp\left[\langle f_{\mathbf{x}}^*(\mathbf{x}), f_{\mathbf{y}}^*(\mathbf{y}), f_{\mathbf{z}}^*(\mathbf{z})\rangle\right] p(\mathbf{y}) d\mathbf{y} \\
&= \langle f_{\mathbf{x}}^*(\mathbf{x}), f_{\mathbf{y}}^*(\mathbf{y}), f_{\mathbf{z}}^*(\mathbf{z}) \rangle + \log p(\mathbf{y}). \tag{30}
\end{aligned}$$

Since the above integral is constant with respect to $\mathbf{y}$, Equation (30) will produce the same rankings for $\mathbf{y}$ as $\log p(\mathbf{y} \mid \mathbf{x}, \mathbf{z})$.

If the marginal distribution of $\mathbf{y}$ is *not* known, then because $f_{\mathbf{x}}^*(\mathbf{x}) \odot f_{\mathbf{z}}^*(\mathbf{z})$ is a sufficient statistic for predicting $\mathbf{y}$ (Theorem 3.3), we could instead use $f_{\mathbf{x}}^*(\mathbf{x}) \odot f_{\mathbf{z}}^*(\mathbf{z})$ to train a simple model to predict any property of $\mathbf{y}$, $s(\mathbf{y})$: $p(s(\mathbf{y}) \mid f_{\mathbf{x}}^*(\mathbf{x}) \odot f_{\mathbf{z}}^*(\mathbf{z}))$.

# I Experiment details

All datasets and code used in this work are publicly available at https://github.com/rajesh-lab/symile.

For all experiments, we use the AdamW optimizer [32]. Following [40], the temperature parameter $\tau$ is directly optimized during training as a multiplicative scalar to avoid the need for separate hyperparameter tuning. Experiments were conducted with 16 CPUs, 200GB of RAM, and a single NVIDIA A100 80GB PCIe GPU.

## I.1 Simulated data: 1D

We fit a model with three affine linear functions that map the binary data $\mathbf{a}, \mathbf{b}, \mathbf{c}$ to representations $\mathbf{r_a}, \mathbf{r_b}, \mathbf{r_c} \in \mathbb{R}^{16}$, respectively. The zero-shot classification task is to predict whether $\mathbf{r_{b=0}}$ or $\mathbf{r_{b=1}}$ is the correct match for a given $\mathbf{r_a}, \mathbf{r_c}$.

## I.2 Simulated data: 5D

The synthetic dataset is drawn according to the following sampling procedure:

$$a_j, b_j \sim \text{Bernoulli}(0.5), \quad i \sim \text{Bernoulli}(\hat{p}), \quad c_j = (a_j \text{ XOR } b_j)^i \cdot a_j^{(1-i)}$$
$$\mathbf{a} = [a_1, \ldots, a_5], \quad \mathbf{b} = [b_1, \ldots, b_5], \quad \mathbf{c} = [c_1, \ldots, c_5].$$

We construct train, val, and test sets of 10K, 1K, and 5K samples, respectively. We fit three affine linear functions that map $\mathbf{a}, \mathbf{b}, \mathbf{c}$ to representations $\mathbf{r_a}, \mathbf{r_b}, \mathbf{r_c} \in \mathbb{R}^{16}$, respectively. These representations are then L2-normalized.

Both Symile and CLIP are trained for 100 epochs using a batch size of 1000, a learning rate of 0.1, and a weight decay of 0.01. The learned temperature parameter $\tau$ is initialized to $-0.3$. The Symile loss is trained with $O(N)$ negative sampling. Checkpoints were saved at the end of every epoch, and the best model was selected based on the lowest validation loss.

## I.3 Symile-M3

**Dataset.** We use images from the ImageNet Large Scale Visual Recognition Challenge (ILSVRC) 2012-2017 train set [45], which we downloaded from Kaggle.[5] The ImageNet train set has 1,281,167 images from 1,000 categories.

We use audio from the Common Voice Corpus [4]. All languages are from versions 16.0 except for English, which is from version 14.0. Each audio clip in the dataset is an MP3 file that consists of a sentence being read aloud. We remove any audio clips that have duration 0.0 seconds. We use the following languages for each version of Symile-M3:

- Symile-M3-2: English, Greek
- Symile-M3-5: English, Greek, Hindi, Japanese, Ukrainian
- Symile-M3-10: Arabic, Chinese, English, Greek, Hindi, Japanese, Korean, Telugu, Thai, Ukrainian

To generate text, we use Google Cloud's Translation API[6] to translate the ImageNet class names into the relevant language. For the ImageNet class names with identical translations across languages (for example, dog breeds), we manually disambiguate so there is no translation overlap. We publicly release all translations to ensure reproducibility.

For each of the three versions of Symile-M3, 10M training, 500K validation, and 500K test samples were generated.

**Training.** Although Symile does not require the use of pre-trained encoders, we use them to accelerate training. For audio, image, and text, we use pre-trained encoders from Whisper [41] (Hugging Face model id `openai/whisper-large-v3`), CLIP [40] (Hugging Face model id `openai/clip-vit-large-patch14`), and XLM-RoBERTa [13] (Hugging Face model id `xlm-roberta-large`), respectively. Audio is downsampled to 16kHz, as expected by Whisper,

---

[5] https://www.kaggle.com/c/imagenet-object-localization-challenge/overview/description

[6] https://cloud.google.com/translate

before being passed to the feature extractor. We freeze the three encoders' parameters except for those in the text encoder's embedding layer and first encoder layer, which are fine-tuned. We train three linear projections to map each encoder's representation to the same 8192-dimensional space, followed by layer normalization.

For each combination of objective (Symile or CLIP) and Symile-M3 version (2, 5, or 10), we do a grid search over learning rate (1e-5, 5e-5, 1e-4) and weight decay (0, 1e-4, 1e-3). We also tune these hyperparameters for the experiments with missing data. All models are trained for 24 epochs using a batch size of 256. The learned temperature parameter $\tau$ is initialized to $-6$. The Symile loss is trained with $O(N)$ negative sampling. Checkpoints were saved every two epochs, and the best model was selected based on the lowest validation loss.

**Missingness.** We evaluate Symile on a variant of Symile-M3-2 where each modality is independently missing with probability $0.5$ or $0.65$, which correspond, respectively, to probabilities $0.125$ and $0.043$ of a complete data sample.

For audio and image data, we learn two embeddings, one for observed data points and one for missing data points. Each embedding matches the dimension of the last hidden layer of the respective audio or image encoder. When a data point is observed, we concatenate its encoder representation and the learned embedding for observed data points, and pass this combined vector into the linear projection head before layer normalization. When a data point is missing, we concatenate the mean encoder representation from the observed training samples and the learned embedding for missing data points, and pass this combined vector into the linear projection head before layer normalization.

For text data, if a data point is missing, we pass into the text encoder the tokenized representation of `[MISSING]`, which is outside of the model's vocabulary.

### I.4 Symile-MIMIC

Symile-MIMIC is a clinical dataset comprised of chest X-rays, electrocardiograms, and blood labs from the MIMIC-IV [16, 17, 24, 27] and MIMIC-CXR [25, 26] datasets. We use admissions and labs from MIMIC-IV v2.2,[7] ECGs from MIMIC-IV-ECG v1.0,[8] and CXRs from MIMIC-CXR-JPG v2.0.0.[9]

Each data sample includes an ECG reading and blood labs taken within 24 hours of the patient's admission to the hospital, and a CXR taken in the 24-72 hour period post-admission. For each admission, we choose the earliest CXR, ECG, and labs.

We use CXRs in JPG format, and consider only CXRs with a posteroanterior (PA) or anteroposterior (AP) view. Following Irvin et al. [23], each CXR is scaled such that the smaller edge is set to 320 pixels, followed by a square crop (random for training or center for validation and testing). Images are then normalized using the ImageNet mean and standard deviation.

We use 10-second 12-lead ECGs, and remove from consideration any ECGs with NaN values or with a signal of all zeros. The ECG signal is normalized to lie within the range $[-1, 1]$.

We focus on the following 50 most common blood laboratory measurements in our dataset, with each data sample containing at least one: Hematocrit, Platelet Count, Creatinine, Potassium, Hemoglobin, White Blood Cells, MCHC, Red Blood Cells, MCV, MCH, RDW, Urea Nitrogen, Sodium, Chloride, Bicarbonate, Anion Gap, Glucose, Magnesium, Calcium Total, Phosphate, INR (PT), PT, PTT, Basophils, Neutrophils, Monocytes, Eosinophils, Lymphocytes, RDW-SD, H, L, I, Alanine Aminotransferase (ALT), Asparate Aminotransferase (AST), Lactate, Alkaline Phosphatase, Bilirubin Total, pH, Albumin, Base Excess, pO2, Calculated Total CO2, pCO2, Absolute Neutrophil Count, Absolute Eosinophil Count, Absolute Monocyte Count, Absolute Basophil Count, Absolute Lymphocyte Count, Creatine Kinase (CK), Immature Granulocytes.

For the labs model, we use a 100-dimensional vector as input: the first 50 coordinates are lab values standardized to percentiles based on the training set's empirical CDF, and the remaining 50 coordinates are binary indicators that denote whether each lab value is missing. When a lab value is unobserved, the mean percentile for that lab is substituted.

---

[7] https://physionet.org/content/mimiciv/2.2/
[8] https://physionet.org/content/mimic-iv-ecg/1.0/
[9] https://physionet.org/content/mimic-cxr-jpg/2.0.0/

Following previous work [8, 22, 29, 30, 57], we use the ResNet-50 and ResNet-18 architectures [20] for the CXR and ECG encoders, respectively, and a three-layer neural network to encode the blood labs. All encoders are trained from scratch, and three linear projections map each encoder's representation to the same 8192-dimensional space.

For Symile and CLIP each, we do a grid search over learning rate (5e-5, 1e-4, 5e-4, 1e-3, 5e-3, 1e-2) and weight decay (1e-3, 1e-2, 1e-1, 2e-1, 5e-1). All models are trained for 80 epochs using a batch size of 280. The learned temperature parameter $\tau$ is initialized to $-7$. The Symile loss is trained with $O(N^2)$ negative sampling to mitigate overfitting. Checkpoints were saved at the end of every epoch, and the best model was selected based on the lowest validation loss.

