# OpenReview forum: "Contrasting with Symile: Simple Model-Agnostic Representation Learning for Unlimited Modalities"
_NeurIPS.cc/2024/Conference — NeurIPS 2024 poster_

### Official Review · Reviewer_hrNA · 2024-07-10

**Soundness:** 3
**Presentation:** 3
**Contribution:** 3
**Rating:** 7
**Confidence:** 3

**Summary:**

This paper shows that CLIP-style pairwise contrastive objectives on multiple modalities (more than two) fail to capture the total dependencies of different modalities, and proposes to use total correlation, a higher-order generalization of mutual information as the new objective to optimize multimodal contrastive learning. The authors provide derivations of the lower bound on total correlation, the optimal scoring function, the practical objective, and proof of statistical sufficiency. On a simulated dataset, a newly created multilingual dataset, and a Chest X-ray dataset, the authors illustrate the limitation of existing CLIP-style pairwise objectives and the effectiveness of the proposed approach.

**Strengths:**

Clarity: the paper is very well written. The motivation, problem formulation, derivation, and novelty of the paper are clearly stated. Figures 1 and 2 are helpful for the reader to understand the concepts.

Quality: the theoretical derivations seem valid to the reviewer upon checking. The theoretical results of the lower bound, optimal scoring function, and statistical sufficiency make this work concrete. The derivations in Appendix A seem correct, and the reviewer did not check the derivations in other Appendix sections. This paper also introduces a new multilingual dataset, Symile-M3, including 33 million (audio, image, text) samples.

Significance: the paper points out that pair-wise CLIP losses cannot capture conditional dependencies between multiple variables, and provides a simple and effective solution to address this. The problem of leveraging contrastive learning on more than two modalities is worth studying.

**Weaknesses:**

Originality: the problem of capturing dependencies of multiple variables beyond pairwise mutual information is well-studied. There are many solutions proposed already, albeit not in an explicitly multimodal setup, as discussed by the authors in Lines 226-231. However, the reviewer reckons that the methods such as Bai et al. can be extended to the setting involving three or more modalities, e.g., substituting the $x^1, ..., x^m$ by different modalities in the Sample-based TC estimator from Bai et al. The authors did not include any such baseline results on the datasets. The reviewer would appreciate clarifications on this.

Quality: the authors did not justify clearly why they curated a new dataset (and a multilingual one) instead of using existing multimodal datasets, e.g., VQA, GQA, How2, HowTo100M, etc. Lacking comparisons on standard datasets weakens the submission.

**Questions:**

There are no error bars in Figures 5 and 6, will the authors intend to provide error bars for them?

There is a quite related paper on capturing total information and conditional independence in a multimodal setup. The reviewer would appreciate some comparisons with it:
Liang, Paul Pu, et al. "Factorized contrastive learning: Going beyond multi-view redundancy." NeurIPS 2023.

**Limitations:**

The authors include limitations of the work.

---

> ### Author Rebuttal · Authors · 2024-08-07
>
> We greatly appreciate your helpful and constructive comments! We are glad that you found our contributions well motivated and clearly explained, and that you consider our solution to be simple and effective. We have addressed your questions regarding suitable baselines for Symile both below and in the global response above, but do let us know if there are any further clarifications we can provide. If we have addressed your main concerns, would you be willing to consider raising your score?
>
> **Bai et al. as a baseline:** While Symile optimizes only a single term in targeting total correlation (TC), Bai et al. derive TC estimators by recursively decomposing TC into a summation of MI terms, to which variational estimators are applied, using two linear separation paths:
> $$\text{Line-like: } TC(X_{1:i+1})=TC(X_{1:i})+I(X_{1:i};x_{i+1})$$
> $$\text{Tree-like: } TC(X_{i:j})=TC(X_{i:\lfloor(i+j)/2\rfloor})+TC(X_{\lfloor(i+j)/2\rfloor+1:j})+I(X_{i:\lfloor(i+j)/2\rfloor};X_{\lfloor(i+j)/2\rfloor+1:j})$$
> The authors apply the tree-like TC estimator to multi-augmentation contrastive learning, where a text encoder is trained by maximizing the TC between augmentations of the text. For this experiment, the authors use InfoNCE to estimate the MI terms and find that their estimator does not perform much better than pairwise InfoNCE.
>
> As we discuss in the global response, Bai et al. and others have leveraged TC estimators for contrastive learning on multiple augmentations of the same data, but to our knowledge no one has applied such estimators to more than two distinct modalities. We also explain in the global response why pairwise CLIP is the most suitable baseline for Symile, but we recognize that we could have provided a better framing for how we situate our work in comparison to Bai et al., and have updated Related Work accordingly.
>
> **Justification for new dataset:** We created the Symile-M3 dataset for two reasons:
> 1. There should exist a dataset specifically designed to evaluate a model's ability to capture higher-order information between modes.
> 2. We wanted a dataset comprised of distinct data types for more than two modes.
>
> By incorporating multiple languages, we were able to design a task where two modes (text and audio) were needed to predict the third (image), and where, importantly, neither text nor audio alone would suffice for predicting image. That said, we agree that it is important to run experiments on real-world datasets, which is why we ran the healthcare experiment on the existing MIMIC dataset in Section 5.3.
>
> We considered video data, but found that we could not use datasets such as How2 and HowTo100M because the text is a direct transcription of the audio. When one mode is a function of another, the dataset effectively has only two modes, and total correlation between three modes degenerates to mutual information between two modes.
>
> While we had not originally considered datasets like VQA because they incorporate only two data types, we agree that including results on standard datasets would strengthen our paper. Based on your suggestion, we ran preliminary experiments using available encoders on the VQA dataset where the three modes are 1) an image, 2) a text question (e.g. "Who is wearing glasses?"), and 3) a text answer (e.g. "woman"). On the task of predicting one of 18 possible answers given the image and question, Symile outperformed CLIP with a mean test accuracy of 0.590 vs. 0.558. We intend to run this experiment more extensively, and will add the findings to our paper.
>
> **Comparison with Liang et al.:** While Liang et al. use contrastive learning objectives to optimize conditional MI terms, their approach is restricted to handling only two modalities.
>
> For modes $X_1$ and $X_2$, Liang et al. propose two contrastive objectives (supervised and self-supervised) that apply lower bounds to maximize task-relevant information and upper bounds to minimize task-irrelevant information. The supervised objective introduces a task $Y$:
> $$
> L_{\text{FactorCL-SUP}}=I_{\text{NCE}}(X_1;X_2)-I_{\text{NCE-CLUB}}(X_1;X_2|Y)+I_{\text{NCE}}(X_1;Y)+I_{\text{NCE}}(X_2;Y)-I_{\text{NCE-CLUB}}(X_1;X_2)+I_{\text{NCE}}(X_1;X_2|Y)
> $$
> where $I_{\text{NCE}}$ is the InfoNCE objective and $I_{\text{NCE-CLUB}}$ is another contrastive objective derived from an upper bound on mutual information that depends on the optimal scoring function from $I_{\text{NCE}}$.
>
> Instead of using explicit task labels $Y$, the self-supervised objective uses augmentations $X_1'$ and $X_2'$ to approximate task-relevant information:
> $$L_{\text{FactorCL-SSL}}=I_{\text{NCE}}(X_1;X_2)-I_{\text{NCE-CLUB}}(X_1;X_2|X_1',X_2')+I_{\text{NCE}}(X_1;X_1')+I_{\text{NCE}}(X_2;X_2')-I_{\text{NCE-CLUB}}(X_1;X_2)+I_{\text{NCE}}(X_1;X_2|X_1',X_2').$$
> Unlike FactorCL, Symile does not require prior knowledge of some specific downstream task defined by either explicit labels or targeted augmentations. FactorCL seeks to decompose the information in two modes for a specific task; Symile targets all information across any number of modes to learn general representations without any decomposition, resulting in computational benefits.
>
> While Symile optimizes a single objective with which all parameters can be updated at once, training either of the FactorCL objectives involves the minimization of each of the $I_{\text{NCE-CLUB}}$ objectives, which in turn requires the optimal critic $f^∗$ from $I_{\text{NCE}}$. Therefore, within each iteration during FactorCL training, one needs to first obtain the optimal critics for the $I_{\text{NCE-CLUB}}$ terms using the $I_{\text{NCE}}$ objective. Symile is able to capture higher-order information for more than two modes while avoiding such computational complexity.
>
> In summary, FactorCL targets the information in two modes for a specific task, and in support of this target makes use of higher-order information. We have added this to Related Work.
>
> **Error bars:** We will provide error bars for all experiments.

---

> ### Comment · Reviewer_hrNA · 2024-08-08
> **Reviewer's response**
>
> The reviewer is fully satisfied with the global and specific responses and increased the score. The reviewer recommends that the authors include the VQA results and the Symile-M3 dataset discussions in the final draft.

---

> > ### Author Response · Authors · 2024-08-10
> >
> > Thank you for taking the time to engage with our responses! We will absolutely include the VQA experiments and the Symile-M3 discussion in the final version of the paper.

---

### Official Review · Reviewer_HxmV · 2024-07-11

**Soundness:** 2
**Presentation:** 3
**Contribution:** 3
**Rating:** 7
**Confidence:** 2

**Summary:**

The paper introduces Symile, a new contrastive learning objective designed to accommodate any number of modalities, addressing the limitations of pairwise CLIP which fails to capture joint and conditional information between modalities. Symile targets total correlation, capturing higher-order dependencies among multiple variables. The paper derives a lower bound on total correlation using a generalized form of inner products and demonstrates that Symile representations form sufficient statistics for the remaining modalities. Experiments on a large multilingual dataset and clinical data show that Symile outperforms pairwise CLIP in cross-modal classification and retrieval tasks, even with missing modalities.

**Strengths:**

1. The paper proposes a model-agnostic contrastive learning objective that captures higher-order dependencies among any number of modalities, addressing limitations of existing pairwise CLIP methods. It effectively identifies and addresses a significant limitation of CLIP, presenting a novel and important contribution to the field.
2. The paper starts with a simple one-dimensional problem to illustrate the failure mode of pairwise CLIP, which aids in understanding and intuition.
3. Theoretical analysis is provided to further support that Symile learns sufficient statistics.

**Weaknesses:**

I wonder if more explanation could be provided on the rationale of the MIP. For example, if we have 3 vectors and consider one coordinate, imagine two cases: (1) the values are x= -1, y= -1, z= 1, whose product is 1, and (2) the values are x= -1, y= -1, z= -1, whose product is -1. In this example, case (2) apparently has higher similarity between the 3 values than case (1), but case (1) ends up with a larger MIP value, which seems undesirable to me.

For the experiments, since I am not very familiar with the state of the art in learning with three or more modalities, I'll leave it to other reviewers to judge whether the benchmarks used and the comparisons are sufficient.

**Questions:**

See the question in Weaknesses.

**Limitations:**

The authors have discussed certain limitations in section 6.

---

> ### Author Rebuttal · Authors · 2024-08-07
>
> Thank you for your thoughtful comments and questions! We're thrilled that you found our work valuable and our examples and theoretical contributions effective. Please let us know if we can provide any further clarifications. If our response has resolved your questions, we would greatly appreciate it if you would consider raising your score.
>
> **Explanation for multilinear inner product (MIP):** We chose the MIP as a scoring function for several reasons. We were drawn to its simplicity: the MIP is one of the simplest possible generalizations of the dot product to more than two modalities in terms of computation. The scoring function also needs to be expressive enough to model any joint statistic, which requires that the vectors be multiplied together. Therefore, the MIP strikes a nice balance between computational simplicity and expressiveness.
>
> We understand your confusion regarding the interpretation of the MIP—the MIP is not a measure of how *geometrically* similar vectors are. Suppose we have three modalities $\mathbf{x}, \mathbf{y}, \mathbf{z}$ whose Symile representations are $\mathbf{r_x}, \mathbf{r_y}, \mathbf{r_z}$. If the MIP of $\mathbf{r_x}, \mathbf{r_y}, \mathbf{r_z}$ is large, then that indicates that $(\mathbf{x}, \mathbf{y}, \mathbf{z})$ has high probability under the joint likelihood. It says nothing about whether or not $\mathbf{r_x}, \mathbf{r_y}, \mathbf{r_z}$ are equal to one another. In other words, the MIP is a measure of similarity imbued by the joint distributition of the modalities.
>
> We're glad you brought this up, and have included this clarification in the paper, as it will benefit our readers.
>
> **Baselines:** You'll see that we addressed the issue of baselines and benchmarks in our global response, and in our responses to the other two reviewers. Please let us know if you have any further questions here that we can address.

---

### Official Review · Reviewer_QTPJ · 2024-07-13

**Soundness:** 2
**Presentation:** 2
**Contribution:** 2
**Rating:** 6
**Confidence:** 4

**Summary:**

This paper proposes Symile, a new contrastive learning objective that addresses the challenge of contrasting multiple modalities simultaneously by considering total correlation. Symile captures the statistical dependence between multiple variables and uses a generalized inner product to derive a lower bound on total correlation. The representations learned by Symile form sufficient statistics for other modalities, outperforming pairwise CLIP in cross-modal classification and retrieval. This is demonstrated through experiments on a multilingual dataset with 33 million samples and a clinical dataset.

=== After response ==
I initially assigned a score of 4, but after reviewing the manuscript further and clarifying some misunderstood details addressed in the rebuttal, I decided to adjust my score. Following a discussion of other reviews, I increased my score to 6 (weak accept).

**Strengths:**

Introducing the limitation of pairwise summation of contrastive losses for multiple modalities is interesting, as demonstrated by both a synthetic example and theoretical justification. Addressing this limitation makes the work more principled. Additionally, handling missing data is crucial for practical scenarios.

**Weaknesses:**

Limited experiments:
- The paper only compares the proposed method with CLIP. More recent works, especially those focused on representation learning with multiple modalities, should be included for a comprehensive comparison. ImageBind, with its publicly available weights, would be considered as a baseline.
- The formulation of the Symile model requires complete triplets (e.g., image, text, audio) for training, which makes it difficult to scale up the training data. The current approach does not accommodate missing values in the triplet elements, which is a significant limitation for practical applications. An extension to handle missing values is needed.

**Questions:**

#1. It's a great idea to include a simple and clear example to show why summing pairwise contrastive losses might not be best for multiple modalities. However, the failure analysis seems very theoretical. In real-world datasets, the assumption that random variables are jointly dependent but pairwise independent might not be true. Can you provide more realistic examples where real-world datasets fit this assumption?

#2. What is the reason for using the coordinate-wise sum of the element-wise product of a set of vectors instead of the simple dot product for the scoring function?

#3. The only baseline used to compare the proposed method is CLIP. More recent works, especially those focused on representation learning with multiple modalities, should also be included. In my opinion, ImageBind would be selected as its weights are publicly available.

#4. The negative sampling with O(N) compared to O(N^2) seems to be a very crude approximation. Does it affect the performance? From a theoretical perspective, can we analyze the difference between the linear and quadratic sampling schemes in terms of the total correlation lower bound?

#5. Minor comments
What do the color shades in Figure 1 represent?
In line 138, the element of Z_n​ should be written in lowercase as z_n.
In related works, the authors mention that contrastive learning has been popularized since the release of CLIP. However, in my opinion, the credit for popularizing contrastive learning should go to SimCLR.

**Limitations:**

The authors have mentioned the limitations, aside from the lack of comparisons. Additionally, in my opinion, there is no potential negative societal impact of this work.

---

> ### Author Rebuttal · Authors · 2024-08-07
>
> We appreciate your detailed feedback, and are glad that you found our work conceptually and theoretically compelling. We address your questions below, but please let us know if we can provide any further clarifications. If we have successfully addressed your primary concerns, would you be willing to consider raising your score?
>
> **ImageBind as a baseline:** As we discuss in the global response above, the baseline that we currently use in our paper is pairwise CLIP, and ImageBind is an instantiation of pairwise CLIP. That said, your feedback indicates that we could have made this clearer in Related Work, which we have now updated.
>
> **Accommodating data missingness:** It seems there was a misunderstanding here. In our missingness experiment in Section 5.2, we do indeed train Symile with missingness in the training data. On the right-hand side of Fig. 5, we show that Symile outperforms pairwise CLIP on the 2-word subset of Symile-M3 when only 12.5% and 2.7% of the training data has complete triplets! We will update this section so that it's clearer to readers that this experiment was run with missingness in the training data.
>
> We absolutely agree that in order for Symile to be at all useful for practical applications, it needs to be able to accomodate any amount of missingness in the data, which is why we ran the missingness experiments in Section 5.2. Our approach for handling missingness was specifically designed to be easy to implement for practical applications. Please let us know if there is additional information we can provide to help clarify this misunderstanding.
>
> **Real-world scenarios where Symile outperforms CLIP:** We're glad that you found the XOR example to be clear and illustrative of the types of information that Symile and pairwise CLIP capture, and you are correct that it represents an extreme case that is probably rare in real-world applications. To see why, consider again the total correlation decomposition in line 115 of our paper:
> $$3 \cdot TC(x,y,z) = 2 \cdot [MI(x;y)+MI(y;z)+MI(x;z)] + MI(x;y | z)+MI(y;z | x)+MI(x;z | y).$$
> In the XOR example, the CLIP target is zero, but as you rightly suggest, most real-world cases will contain *some* pairwise information.
>
> Based on the decomposition above, Symile will outperform CLIP in situations where the higher-order (conditional) information terms play a role. This is typically seen in applications when two modes are needed to infer the third.
>
> We include one such example in Section 5.3 of our paper on the task of predicting chest X-ray from ECG and labs: there is pairwise information to learn, which explains why CLIP performs better than random guessing, but the presence of conditional information allows Symile to ultimately outperform CLIP. As suggested by Reviewer hrNA, we ran preliminary experiments on the VQA dataset and found that Symile outperforms CLIP, illustrating another instance where conditional information benefits Symile.
>
> **Multilinear inner product (Q2):** The dot product is only defined for two modes. The coordinate-wise sum of the element-wise product is one of the simplest possible generalizations of the dot product to more than two modes. Please let us know if we misunderstood the question or can clarify further.
>
> **Negative sampling performance impact and lower bound connection:** The results we report in the paper on the Symile-M3 dataset were run using $O(N)$ negative sampling. We have now also run this experiment on the 2-word subset of Symile-M3 using $O(N^2)$ negative sampling. We find that when trained using $O(N^2)$ negative sampling, Symile demonstrates a marginal improvement in test accuracy over training with $O(N)$ negative sampling (0.938 vs. 0.937). However, Symile trained with $O(N^2)$ negative sampling takes almost twice as long to run 24 epochs (43 vs. 24 hours), but achieves its best validation accuracy in far fewer epochs (4 vs. 16). We have added these results to the Appendix.
>
> As discussed in Section 5.3, we found that training with $O(N^2)$ negative sampling on the Symile-MIMIC dataset helped mitigate overfitting. We hypothesize that this is because more negative samples create a more challenging learning problem, allowing the model to better learn to differentiate between positive and negative samples.
>
> In terms of analyzing the difference between the negative sampling schemes in terms of the derived lower bound, in lines 156-157 we write, "We show in Appendix B.3 that, as $N$ gets larger, the total correlation lower bound closes for the optimal scoring function $g^*$." This implies a computational-statistical trade-off: a larger batch size demands more computation but results in a tighter bound.
>
> **Colors in Fig. 1:** The colors are meant to loosely represent the information in each of the three random variables. We carefully considered how best to structure this figure: on the one hand, we wanted a high-level and intuitive illustration of the different types of information that Symile and CLIP each capture (and it seems that Reviewer hrNA appreciated this); on the other hand, we understand that this figure is underspecified. Ultimately, we decided to err on the side of building the reader's intuition.
>
> **SimCLR vs. CLIP in Related Work:** We write in Related Work that CLIP "popularized the use of a contrastive objective for general image-text representation learning," but we agree with you that SimCLR popularized contrastive learning for multiple augmentations for a single modality. We will update this section accordingly.
>
> **$Z_n$ typo:** Thank you for catching this!

---

### Author Rebuttal · Authors · 2024-08-07

We thank the reviewers for their constructive and careful comments, which will significantly strengthen our work. We are glad they found that:
* our paper "effectively identifies and addresses a significant limitation of CLIP, presenting a novel and important contribution to the field" (HxmV)
* our work "provides a simple and effective solution to address this [limitation]" (hrNA)
* "the problem of leveraging contrastive learning on more than two modalities is worth studying" (hrNA) and "[a]ddressing this limitation makes the work more principled" (QTPJ)
* "The paper is very well written. The motivation, problem formulation, derivation, and novelty of the paper are clearly stated." (hrNA)

### Common reviewer feedback

A shared point that arose concerned appropriate baselines for Symile. We sincerely appreciated these questions—they helped to refine our presentation of Symile's contributions to representation learning. We discuss the baselines that came up below.

**ImageBind**

Reviewer QTPJ writes, "ImageBind, with its publicly available weights, would be considered as a baseline." The baseline that we currently use in our paper is pairwise CLIP, and ImageBind is pairwise CLIP. To see why, let $I$ represent images with learned embeddings $q$ and let $M_\ell$ represent one of the other five modalities with learned embeddings $k_\ell$. Notice that ImageBind uses the InfoNCE loss:
$L_{\mathcal{I},\mathcal{M_\ell}} = -\log \frac{\exp{(q_i^\top k_{\ell,i}} / \tau)}{\exp{(q_i^\top k_{\ell, i}} / \tau) + \sum_{j \neq i} \exp{(q_i^\top k_{\ell, j}} / \tau)}.$
ImageBind then trains by minimizing this loss summed across the other modes indexed by $\ell$. *This is pairwise CLIP where only pairs where the image mode appears are kept.*

In our experiments, we compare Symile to pairwise CLIP—the ImageBind approach—and find that Symile outperforms CLIP.

**Contrastive representation learning on multiple augmentations**

Reviewer hrNA rightly points out that previous work has explored contrastive learning in the context of multiple views of the same data. For example, Bai et al. (2023) derive two total correlation (TC) estimators, one of which they apply to multi-augmentation contrastive learning where a text encoder is trained by maximizing the TC between four augmentations of the text (see our individual response to hrNA below for details). Shidani et al. (2024), which we mention in Related Work, develop a (pairwise) contrastive approach using more than two augmentations for image representation learning. Reviewer hrNA also mentioned Liang et al. (2023): while they use contrastive learning objectives to optimize conditional MI terms, their approach is restricted to only two modalities.

The relationship between our work and contrastive methods for multi-augmentations on a single modality is comparable to that between CLIP (Radford et al., 2021) and SimCLR (Chen et al., 2020). While SimCLR popularized the mutual information (MI) estimator InfoNCE (van den Oord et al., 2018) for contrastive learning on two augmentations of the same data, CLIP used this estimator to handle distinct modalities. Similarly, Bai et al. (and many others before them) leverage TC or MI estimators for contrastive learning on multiple augmentations of the same data, but to our knowledge no one has made use of such estimators for more than two distinct modalities—except for pairwise CLIP, which we do use as a baseline. Further, these estimators typically require specific encoders that ingest multiple modes of data, which limits their use.

The contribution of our paper is a combination of the contributions of InfoNCE and CLIP for more than two modes of data. Our paper (1) provides a single contrastive loss that works with any encoders and recovers higher-order information for more than two modes (like InfoNCE for two modes) and (2) demonstrates the value of this estimator for representation learning on more than two distinct modes of data (like CLIP for two modes).

Given our work's emphasis on targeting TC for representation learning for distinct modalities using any type of encoder, we view pairwise CLIP as the most suitable baseline because pairwise CLIP is intended for distinct data modalities and allows for the use of any encoders for those modalities. We also show why targeting TC yields good representations by, for example, showing that Symile yields sufficient statistics; this analysis holds for any TC estimator that at optimality yields a likelihood ratio. That said, it is certainly an interesting and worthwhile question to ask how our approach compares to other TC estimators—but it is outside of the scope of our paper.

### Contributions
We now summarize the contibutions of our work.

**Theoretical:**
- Show that, despite its popularity, the pairwise use of CLIP fails to capture higher-order information between modalities, thereby limiting the quality of the representations it learns
- Propose the use of total correlation for representation learning with more than two modalities of data by noting it captures higher-order information
- Derive a multi-sample lower bound on total correlation to build Symile, a simple contrastive learning objective that, unlike other total correlation estimators, accommodates any number of modalities and allows any model to produce representations for each modality
- Prove that representations produced by Symile for any set of modalities form sufficient statistics for the remaining modalities not considered in the set

**Empirical:**
- Demonstrate that Symile outperforms pairwise CLIP on cross-modal classification and retrieval in a multilingual dataset of images, text and audio of over 33M examples and a clinical dataset of chest X-rays, electrocardiograms, and laboratory measurements
- Show that Symile retains its advantage over pairwise CLIP even with modalities missing in the data

---

### Decision · Program_Chairs · 2024-09-25

**Decision:**

Accept (poster)

**Comment:**

This paper introduces Symile, a novel contrastive learning objective designed to capture higher-order dependencies among multiple modalities, addressing limitations of pairwise CLIP approaches. The authors provide theoretical foundations, deriving a lower bound on total correlation and demonstrating that Symile representations form sufficient statistics for remaining modalities.

Given the paper's novel contributions, strong theoretical foundations, and comprehensive empirical validation, along with the authors' thorough addressing of reviewer concerns, I recommend accepting this paper for publication.